# Transcriptome and Physiological Analyses of a Navel Orange Mutant with Improved Drought Tolerance and Water Use Efficiency Caused by Increases of Cuticular Wax Accumulation and ROS Scavenging Capacity

**DOI:** 10.3390/ijms23105660

**Published:** 2022-05-18

**Authors:** Beibei Liang, Shiguo Wan, Qingling Ma, Li Yang, Wei Hu, Liuqing Kuang, Jingheng Xie, Dechun Liu, Yong Liu

**Affiliations:** Department of Pomology, College of Agronomy, Jiangxi Agricultural University, Nanchang 330045, China; liang2133533@163.com (B.L.); wsg0045672@163.com (S.W.); mql805472509@163.com (Q.M.); yangli526526@126.com (L.Y.); wei.hu1986@foxmail.com (W.H.); kuangliuqing@jxau.edu.cn (L.K.); 18720986781@163.com (J.X.)

**Keywords:** navel orange mutant, cuticular waxes, drought stress, water use efficiency, ROS scavenging, transcriptome analysis

## Abstract

Drought is one of the main abiotic stresses limiting the quality and yield of citrus. Cuticular waxes play an important role in regulating plant drought tolerance and water use efficiency (WUE). However, the contribution of cuticular waxes to drought tolerance, WUE and the underlying molecular mechanism is still largely unknown in citrus. ‘Longhuihong’ (MT) is a bud mutant of ‘Newhall’ navel orange with curly and bright leaves. In this study, significant increases in the amounts of total waxes and aliphatic wax compounds, including n-alkanes, n-primary alcohols and n-aldehydes, were overserved in MT leaves, which led to the decrease in cuticular permeability and finally resulted in the improvements in drought tolerance and WUE. Compared to WT leaves, MT leaves possessed much lower contents of malondialdehyde (MDA) and hydrogen peroxide (H_2_O_2_), significantly higher levels of proline and soluble sugar, and enhanced superoxide dismutase (SOD), catalase (CAT) and peroxidase (POD) activities under drought stress, which might reduce reactive oxygen species (ROS) damage, improve osmotic regulation and cell membrane stability, and finally, enhance MT tolerance to drought stress. Transcriptome sequencing results showed that seven structural genes were involved in wax biosynthesis and export, MAPK cascade, and ROS scavenging, and seven genes encoding transcription factors might play an important role in promoting cuticular wax accumulation, improving drought tolerance and WUE in MT plants. Our results not only confirmed the important role of cuticular waxes in regulating citrus drought resistance and WUE but also provided various candidate genes for improving citrus drought tolerance and WUE.

## 1. Introduction

Drought stress severely limits plant growth and development and reduces crop yield and quality all over the world [1]. In plants, cuticular waxes are primarily composed of very-long-chain fatty acids (VLCFAs) and their derivatives, including aldehydes, alkanes, alcohols, ketones and esters. In addition, cyclic compounds, such as terpenoids, flavonoids and sterols, are also present in the cuticular waxes of many plant species [2]. As a hydrophobic barrier, cuticular waxes cover the plant surfaces to prevent non-stomatal water loss and protect plants against abiotic and biotic stresses such as UV radiation, cold, drought, high salt, pathogens and insect invasion [3].

Previous reports revealed that cuticular waxes play an important role in regulating plant tolerance to drought stress. For example, the crops with high wax contents were more tolerant to drought stress than those crops containing low wax contents [4]. The overexpression of Arabidopsis *AtMYB96* and apple *MdLACS2* increased the accumulation of cuticular waxes and enhanced the drought tolerance in transgenic plants [5,6]. Silenced expression of rice *OsGL1-6* significantly decreased the total wax content and reduced the plant resistance to drought [7]. Moreover, it has been reported that cuticular waxes are positively related to water use efficiency (WUE) in plants [8,9]. In general, an increased WUE can improve crop productivity and reduce water use under drought stress [10]. Therefore, the relationship between cuticular waxes and WUE should be studied on more crop species.

Citrus is a perennial fruit crop with a long orchard life and is often exposed to drought stress under field conditions. Drought stress is known to limit citrus growth and reduces its fruit quality and economic value [11]. Thus, it is very important to study the mechanisms of citrus response to drought stress. Previous reports revealed that a series of changes occurred in citrus under drought stress, such as physiological changes (stomatal closure, reduction in leaf water potential and gas exchange, and hydraulic redistribution), biochemical changes (production of osmolytes, increases in antioxidant enzyme activities, changes in amino acid metabolism and synthesis of hormones), as well as molecular changes (signal transduction, gene expression changes and DNA methylation) [12,13,14,15]. Cuticular waxes play an important role in the plant’s response to drought stress [3]. Like other plants, the surfaces of citrus aerial organs were covered with a cuticular wax layer. The chemical composition of cuticular waxes has been studied in many citrus species, such as lemon, ‘Clementine’ mandarin, ‘Willowleaf’ mandarin and ‘Valencia’ orange [16,17], grapefruit [18], sweet orange [19], Satsuma mandarin [20] and navel orange [21,22,23,24]. Further study revealed that the water loss in citrus fruits was obviously increased after the removal of cuticular waxes [25]. Our recent report showed that the water loss of navel orange during cold storage were regulated by total wax content and wax fractions such as n-alkanes, n-fatty acids, n-aldehydes, and terpenoids [26]. These results showed that cuticular waxes played an important role in regulating water loss in citrus fruits. The water loss is closely related to plant drought tolerance [27]. Thus, the cuticular waxes may make a great contribution to enhancing drought tolerance in citrus. In fact, the important contributions of citrus wax biosynthesis genes to plant drought tolerance were reported in recent studies. For example, overexpression of navel orange *CsKCS6* and *CsMYB96* increased the production of cuticular wax components and enhanced the drought tolerance of transgenic plants [28,29]. However, the contribution of the cuticular waxes of citrus leaves to drought tolerance, WUE and the underlying molecular mechanism is still largely unknown due to the lack of wax-related mutants. ‘Longhuihong’ navel orange (MT) is a spontaneous bud mutant derived from ‘Newhall’ navel orange (*Citrus sinensis* Osbeck cv. Newhall, WT). This variety was found in an orchard in the Nankang area, Ganzhou city, Jiangxi province of China after a severe freezing disaster in 2000. The most striking phenotype of the MT is the curly (curl to abaxial surface) and bright leaves with prominent veins [30]. The high brightness on the surfaces of plant organs usually indicates the changes in the morphology and chemical composition of cuticular waxes [2,3]. The cuticular waxes make a great contribution to plant drought tolerance [4,5,6,7]. Therefore, the MT provides a valuable mutant material for studying the relationship between cuticular waxes and citrus drought tolerance. In this study, the physiological indexes and WUE under control and drought conditions were assessed in WT and MT plants to determine the drought tolerances of these two varieties. To understand the contribution of cuticular waxes to drought tolerance and WUE in citrus, the differences in morphology and chemical composition of cuticular waxes were also compared between WT and MT leaves. Furthermore, transcriptomic and real-time quantitative PCR (qRT-PCR) analyses were performed to identify candidate genes potentially increasing cuticular wax accumulation and improving drought tolerance and WUE in citrus.

## 2. Results

### 2.1. Comparison of Phenotype, Chromatic Aberration and Cuticular Permeability between WT and MT Leaves

Chromatic aberration analysis revealed that MT leaves had a significantly lower *a** value and much higher *b**, *a**/*b** and *L** values than those in WT leaves, suggesting the MT leaves were much greener and brighter than the WT leaves (Figure 1D–G). However, the two varieties shared similar fresh and dry weight in leaves (Figure 1H,I). Interestingly, the water loss rates in MT leaves were significantly lower than those in WT leaves from 8 to 48 h of dehydration (Figure 1J). Thus, the MT leaves rolled much more severely than WT leaves after 48 h of dehydration (Figure 1A,B). Moreover, the chlorophyll leaching rates in MT leaves were much lower than those in WT leaves from 2 to 30 h after alcohol treatment (Figure 1K). Therefore, the alcohol solution of MT leaves was much lighter green than that of WT leaves after 30 h of alcohol treatment (Figure 1C). These results suggested that the cuticular permeability in MT leaves was much lower than that in WT leaves.

### 2.2. Comparison of Cuticular Wax Morphology and Chemical Composition between WT and MT Leaves

The scanning electron microscopy (SEM) results showed that the epidermic cells were obviously convex on the adaxial surfaces of MT leaves (Figure 2A,B,E,F). On the adaxial and abaxial sides, the wax crystal density in MT leaves was higher than that in WT leaves (Figure 2A–H).

Compared to WT leaves (6.40 μg·cm^−2^), the total wax content increased by 50.78% in MT leaves (9.65 μg·cm^−2^) (Figure 3A). The most abundant wax fraction in WT leaves was n-primary alcohols (3.85 μg·cm^−2^, accounted for 60.16% of total wax content), followed by triterpenoids (1.21 μg·cm^−2^, 18.91%), n-alkanes (0.59 μg·cm^−2^, 9.22%), n-fatty acids (0.48 μg·cm^−2^, 7.50%), sterols (0.19 μg·cm^−2^, 2.97%), unsaturated fatty acids (0.072 μg·cm^−2^, 1.13%) and n-aldehydes (0.0027 μg·cm^−2^, 0.042%). In MT leaves, the most abundant wax fraction was also n-primary alcohols (6.07 μg·cm^−2^, 62.90%), followed by n-alkanes (1.83 μg·cm^−2^, 18.96%), triterpenoids (1.20 μg·cm^−2^, 12.44%), n-fatty acids (0.40 μg·cm^−2^, 4.15%), sterols (0.081 μg·cm^−2^, 0.84%), unsaturated fatty acids (0.075 μg·cm^−2^, 0.78%) and n-aldehydes (0.0034 μg·cm^−2^, 0.035%). The amounts of n-alkanes, n-primary alcohols, and n-aldehydes in MT leaves increased by 210.17%, 57.66% and 25.93%, respectively, compared with those in WT leaves. On the contrary, the amounts of sterols decreased by 57.37% in MT leaves in comparison to WT leaves (Figure 3B).

At the individual level, a total of 22 cuticular wax constituents were detected in WT and MT leaves. There were no significant differences in the amounts of C16:0, C18:0, C20:0 and C24:0 fatty acids, C18:2 and C18:3 unsaturated fatty acids, C23:0, C25:0 and C27:0 alkanes, C34:0 primary alcohol, α-Amyrin and β-Amyrin between WT and MT leaves. However, the amounts of three n-alkane constituents with odd-numbered chain lengths from C29 to C33, five n-primary alcohol constituents with even-numbered chain lengths from C24 to C32 and one n-aldehyde constituent (C26:0 aldehyde) in MT leaves were significantly higher than those in WT leaves. On the contrary, the campesterol deposited on the surfaces of MT leaves with much lower amounts in comparison to WT leaves (Figure 3C).

### 2.3. Comparison of Morphological and Physiological Responses of WT and MT Plants to Drought Stress

The WT and MT plants grew well under control conditions (Figure 4A,B). After 21 days of drought treatment, severely wilting was observed in almost all of the WT leaves (Figure 4C). In contrast, light wilting occurred only in a few leaves of MT plants (Figure 4D). Further study revealed that almost all physiological indexes increased after drought treatment, except for chlorophyll content, which decreased under drought stress (Figure 4E–M). The ion leakage, malondialdehyde (MDA) and hydrogen peroxide (H_2_O_2_) contents in MT leaves were significantly lower than those in WT leaves under control and drought conditions (Figure 4E,F,J). In contrast, the MT leaves exhibited much higher contents of proline, soluble sugar, chlorophyll and significantly increased superoxide dismutase (SOD), peroxidase (POD) and catalase (CAT) activities in comparison with WT leaves under control and drought conditions (Figure 4G–I,K–M).

Photosynthetic analysis showed that the net photosynthetic rate (Pn), stomatal conductance (Gs), intercellular CO_2_ concentration (Ci) and transpiration rate (Tr) decreased after drought treatment in WT and MT leaves (Figure 5A–D). The Pn in MT leaves was significantly higher than that in WT leaves under control and drought conditions (Figure 5A). However, the Gs, Ci and Tr in MT leaves were much lower than those in WT leaves under control conditions and drought stress (Figure 5B–D). Compared to WT plants, the MT plants possessed much higher instantaneous water use efficiency (WUEi) and δ^13^C under control and drought conditions (Figure 5E,F). These results suggested that MT plants had enhanced drought tolerance and increased water use efficiency (WUE) in comparison to WT plants.

### 2.4. Functional Classification of Differentially Expressed Genes (DEGs) between WT and MT Leaves

To investigate the molecular mechanism leading to the phenotypic and wax com-position differences between WT and MT leaves, total RNAs from WT and MT leaves were sequenced with three biological replicates per variety. According to the criteria of |log_2_ (MT/WT)| ≥ 1 and Q-value < 0.001, a total of 1316 DEGs were identified in MT vs. WT, of which 537 DEGs were upregulated and 779 DEGs were downregulated in MT leaves (Figure 6A). The information on all DEGs are listed in Appendix A. Furthermore, a volcano plot of DEGs was constructed to visualize the distribution of upregulated and downregulated DEGs (Figure 6B).

The Gene Ontology (GO) and Kyoto Encyclopedia of Genes and Genomes (KEGG) pathway analyses were used to explore the potential functions of the DEGs. The GO classification analysis showed that a total of 971 DEGs were assigned to the 1513 GO term under three categories, including biological process, molecular function and cellular component. In these GO terms, catalytic activity had the largest number of DEGs (517), followed by binding (465), membrane (392), membrane part (386), cellular process (189), metabolic process (177) and other GO terms (Figure 7A and Appendix A).

According to the Q-value, the most enrichment GO term was oxidoreductase activity, followed by drug catabolic process, heme binding, tetrapyrrole binding, oxidoreductase activity, acting on paired donors, with incorporation or reduction in molecular oxygen, monooxygenase activity, cofactor binding, reactive oxygen species metabolic process and other GO terms (Figure 7B and Appendix A). Most of the enriched GO terms were involved in plant response to drought and other stresses. 

KEGG classification analysis revealed that a total of 645 DEGs were assigned to 120 pathways under five categories, including cellular processes, environmental information processing, genetic information processing, metabolism and organismal systems. The greatest number of DEGs belonged to global and overview maps (324), followed by biosynthesis of other secondary metabolites (123), environmental adaptation (114), signal transduction (111), carbohydrate metabolism (101), lipid metabolism (65), amino acid metabolism (52) and other pathways (Figure 7C and Appendix A). Most of these pathways were related to plant response to drought stress or other stresses, suggesting these DEGs might contribute to enhancing MT drought tolerance.

KEGG enrichment analysis was performed for the criteria of Q-value ≥ 0.05. According to the Q-value, the mitogen-activated protein kinase (MAPK) signaling pathway plant was the most enriching pathway, followed by phenylpropanoid biosynthesis, flavonoid biosynthesis, brassinosteroid biosynthesis, ABC transporters, cutin, suberine and wax biosynthesis, stilbenoid, diarylheptanoid and gingerol biosynthesis, plant-pathogen interaction and other pathways (Figure 7D and Appendix A). Most of these pathways were involved in the plant’s response to environmental stress and cuticular wax biosynthesis, which might be the major reason for the difference in drought resistance between WT and MT plants.

### 2.5. Identification of DEGs Involved in Cuticular Wax Biosynthesis and Transport

A total of eight DEGs (three upregulated and five downregulated) involved in wax biosynthesis and transport were identified in MT vs. WT. The expression levels of *CsCER3-LIKE* encoding very-long-chain aldehyde reductase (CER3), *CsABCG11-LIKE* encoding ATP-binding cassette, subfamily G, member 11 (ABCG11) and *CsABCG21-LIKE* encoding ATP-binding cassette, subfamily G, member 21 (ABCG21) were significantly upregulated in MT leaves. In contrast, the significantly downregulated expression of three *CsFAR-LIKE* genes (*CsFAR1-LIKE*, *CsFAR3-LIKE* and *CsFAR4-LIKE*) encoding fatty acyl-CoA reductases (FARs), *CsNSDHL-LIKE* encoding sterol-4alpha-carboxylate 3-dehydrogenase (NSDHL) and *CsChDI-LIKE* encoding cholestenol delta-isomerase (ChDI) were observed in MT leaves (Table 1).

### 2.6. Identification of DEGs Involved in MAPK Cascade, Reactive Oxygen Species (ROS) Scavenging and Drought Response

According to the Q-value, the MAPK signaling pathway plant was the most significant enrichment pathway of DEGs (Figure 7D), indicating it played important roles in the phenotypic differences between WT and MT. A total of 86 DEGs (50 upregulated and 36 downregulated) were functioning in the MAPK signaling pathway plant (Figure 8A and Appendix A). The KEGG classification analysis showed that 86 DEGs in the MAPK signaling pathway plant were classified as signal transduction, 65 DEGs were involved in environmental adaption and 25 DEGs belonged to the metabolism of stress-related chemical compounds, such as lipid, carbohydrate, amino acid, terpenoids and polyketides (Figure 9A and Appendix A). 

It should be noted that most of the remaining enrichment pathways were biosynthetic, metabolic and signal transduction pathways related to abiotic and biotic stress, such as phenylpropanoid biosynthesis, flavonoid biosynthesis, brassinosteroid biosynthesis, stilbenoid, diarylheptanoid and gingerol biosynthesis, and plant–pathogen interaction, etc. Notably, the plant–pathogen interaction pathway contained the largest number of DEGs (99), suggesting the biotic tolerance of MT plants might also be changed in comparison to WT plants (Figure 7D and Appendix A). In addition, the expression levels of *CsMEKK1-LIKE* encoding MAP kinase kinase kinase (MEKK1), *CsSOD1-LIKE* encoding superoxide dismutase 1 (SOD1) and four *CsPRX-LIKE* genes (*CsPRX5-LIKE*, *CsPRX10-LIKE*, *CsPRX24-LIKE* and *CsPRX25-LIKE*) encoding peroxidases (PRXs) were upregulated in MT leaves, indicating the DEGs involved in the MAPK cascade and ROS scavenging might play an important role in regulating MT tolerance to drought stress (Table 1).

### 2.7. Identification of DEGs Encoding Transcription Factor

A total of 74 DEGs encoding transcription factors were identified in MT vs. WT, of which 30 DEGs were upregulated and 44 DEGs were downregulated in MT leaves. These DEGs belonged to 24 transcription factor families. AP2/ERF and WRKY families were the top two largest transcription factor families, and each family included 10 DEGs. Interestingly, most AP2/ERF DEGs were upregulated in MT leaves. In contrast, most DEGs in the WRKY family were downregulated in MT leaves. Most of the remaining transcription factor families were involved in the plant’s response to drought stress or other stresses, such as NAC, MYB, and WRKY, AP2/ERF, bZIP, bHLH, C3H, MADS, GRAS and Dof (Figure 8B and Appendix A). The KEGG classification analysis revealed that 17 DEGs encoding transcription factors were involved in signal transduction, 15 DEGs were classified to environmental adaption and 7 DEGs were related to the metabolism of stress-related chemical compounds, such as cofactors, vitamins, lipids, amino acids, carbohydrates and glycans (Figure 9B and Appendix A). The upregulated DEGs, including *CsERF4-LIKE*, *CsERF9-LIKE*, *CsMYB62-LIKE*, *CsZAT10-LIKE1*, *CsZAT10-LIKE2*, and *CsNAC22-LIKE,* and downregulated DEGs, such as *CsWRKY27-LIKE* and *CsWRKY29-LIKE,* might make a great contribution to enhance MT resistance to drought stress (Table 1, Figure 8B and Appendix A).

### 2.8. Expression Analysis of Wax Biosynthesis, Transport and Drought Responsive DEGs by qRT-PCR

To validate the results of the transcriptome sequence and investigate the response of DEGs to drought stress, the expression levels of 20 DEGs involved in cuticular wax biosynthesis, transport and drought response were assessed by qRT-PCR under control and drought conditions. The expression patterns of all selected DEGs revealed by qRT-PCR were the same as those of the transcriptome data, demonstrating the reliability of the transcriptome results. The expression levels of 18 DEGs increased under drought stress. However, the *CsWRKY27-LIKE* and *CsWRKY29-LIKE* expression decreased after drought treatment, suggesting they might negatively regulate the navel orange response to drought stress (Figure 10A–T). Compared to WT leaves, significant increases were observed in the transcript levels of *CsCER3-LIKE*, *CsABCG11-LIKE*, *CsABCG21-LIKE*, *CsMEKK1-LIKE*, *CsSOD1-LIKE*, *CsPRX5-LIKE*, *CsPRX10-LIKE*, *CsERF4-LIKE*, *CsERF9-LIKE*, *CsMYB62-LIKE*, *CsZAT10-LIKE1*, *CsZAT10-LIKE2* and *CsNAC22-LIKE* under control and drought conditions (Figure 10A,G–R). In contrast, the expression levels of *CsFAR1-LIKE*, *CsFAR3-LIKE*, *CsFAR4-LIKE*, *CsNSDHL-LIKE*, *CsChDI-LIKE*, *CsWRKY27-LIKE* and *CsWRKY29-LIKE* in MT leaves were much lower than those in WT leaves under control and drought conditions (Figure 10B–F,S,T).

## 3. Discussion

### 3.1. The Decrease in Cuticular Permeability was Caused by the Increase in Aliphatic Wax Accumulation in MT Leaves

The MT tree is a bud mutation derived from the WT tree. A previous report showed that the MT tree possessed curly and bright leaves with prominent veins [30]. In agreement with a previous report, our study revealed that MT leaves were much brighter and greener than WT leaves (Figure 1A,D–G). Furthermore, the water loss rates and chlorophyll leaching rates in MT leaves were significantly lower than those in WT leaves, suggesting the cuticular permeability of MT leaves decreased significantly in comparison to WT leaves (Figure 1B,C,J,K). Further study showed that the MT leaves possessed much higher amounts of total waxes, n-primary alcohols, n-alkanes and n-aldehydes, but significantly lower amounts of sterols in comparison to WT leaves (Figure 3A,B). Previous reports revealed that the formation of epicuticular wax crystals on the surface of citrus was dependent on a high proportion of aliphatic wax compounds such as n-alkanes, n-aldehydes and n-primary alcohols [22,25]. Thus, the significant raises in the amounts of n-primary alcohols, n-alkanes and n-aldehydes could explain the increase in epicuticular wax crystal density on the surfaces of MT leaves (Figure 2A–H). The cuticular waxes are multiphase systems consisting of highly ordered crystalline zones and mobile amorphous domains. The impermeable wax crystalline zones, which contribute greatly to forming the barrier, are mainly made up of aliphatic wax compounds. Water and other solutes can only pass through the amorphous domains, which are composed mainly of pentacyclic components such as sterols and triterpenoids [31]. The significant increase in total wax load in MT leaves, especially the significant increases in n-primary alcohols, n-alkanes and n-aldehydes, fits well with the previous model that indicated aliphatic wax compounds as critical determinants of cuticular permeability [32,33]. Therefore, we deduced that the increases in amounts of aliphatic wax compounds, including n-primary alcohols, n-alkanes and n-aldehydes, might be the major reasons for the reduction in cuticular permeability in MT leaves.

### 3.2. Increased Cuticular Wax Accumulation and Enhanced ROS Scavenging Capacity Contributes to the Improvement of Drought Tolerance and WUE in MT Plants

To investigate whether the increase in cuticular wax accumulation leads to the improvement of drought tolerance in MT plants, the drought tolerance between WT and MT plants was compared in this study. As expected, MT plants were much more tolerant to drought stress than WT plants, as indicated by their fewer wilting leaves under drought stress (Figure 4A–D). This result supported previous reports, which suggested that high cuticular wax deposition could improve plant tolerance to drought stress [5,28,34,35]. Proline and soluble sugar acted as compatible osmolytes involved in plant response to drought stress [36,37]. The increases of proline and soluble sugar contents in MT leaves could keep the osmotic balance between the intracellular and extracellular environments, thus decreasing cellular membrane damage and resulting in enhanced drought tolerance in MT plants (Figure 4G,H). As a soluble product of membrane lipid peroxidation, MDA content is used to assess the extent of lipid peroxidation [38]. Ion leakage is an important indicator of the cellular membrane injury caused by redundant lipid peroxidation [39]. In addition, the increase in H_2_O_2_ content usually causes severe oxidative injury in plant cellular membranes [40]. Therefore, the significant declines in ion leakage rate, MDA levels and H_2_O_2_ content suggested that MT plants suffered less oxidative injury than WT plants under drought stress (Figure 4E,F,J). ROS are highly reactive molecules that can interact with DNA, RNA, proteins, pigments, lipids and numerous other metabolites in plants. Overproduction of ROS in response to drought stress causes serious oxidative damage to plant cells. The antioxidant enzymes involved in the ROS scavenging system reduce oxidative injury and, thus, play an important role in plant tolerance to drought stress [41]. In this study, significantly higher SOD, POD and CAT activities were observed in MT plants compared to WT plants (Figure 4K–M). The enhanced antioxidant enzyme activities might be the major reason for the decline in ROS content (indicated by H_2_O_2_ content) in MT plants under drought stress. All above, we concluded that the enhanced drought tolerance of MT plants could be attributed to two major reasons. The first one was the decrease in cuticular permeability, which was caused by the increase in aliphatic wax accumulation in MT leaves. Another explanation was the improved ROS scavenging capacity, which limited ROS damage and enhanced cell membrane stability in MT leaves.

Drought stress usually causes stomatal closure and consequently decreases Gs and Tr in leaves to limit water loss. The stomatal closure also leads to a decline in the diffusion of CO**_2_**, resulting in a decrease in Pn under drought stress [42]. The Ci was also reported to decline after drought treatment [43]. In agreement with previous reports, the present study showed that the Pn, Gs, Ci and Tr in leaves of both varieties declined after drought treatment. In addition, MT leaves possessed much higher Pn and significantly lower Gs and Ci compared to WT leaves under control and drought conditions (Figure 5A–C). Chlorophyll is the main photosynthetic pigment in plants. A stable supply of chlorophyll is required for plant photosynthesis [44]. Thus, the increased levels of Pn in MT leaves might be attributed to the increase in chlorophyll contents under control and drought conditions. Moreover, the Tr in MT leaves was significantly lower than that in WT under control and drought conditions (Figure 5D). Under drought stress, most of the leaf stomata close and cuticular transpiration becomes the major route for water loss [45]. Thus, the decrease in cuticular permeability caused by an increase in cuticular wax accumulation in MT leaves was probably one of the major reasons for the decline of Tr in MT leaves.

Plant WUE describes the ratio of CO_2_ gain or dry matter production per unit of water loss [46]. High WUE has been reported to improve plant growth, survival and vegetation productivity and reduce water use under drought stress [10]. Plant WUE is often investigated on an instantaneous scale (minutes, WUEi), which is calculated by the ratio of Pn to Tr in leaves [47]. Moreover, leaf δ^13^C has long been used to assess whole plant WUE on short-term (hours or days) and long-term (years or decades) scales since a significant positive correlation has been observed between WUE and δ^13^C in plants [48]. Our results showed that MT plants exhibited much higher WUE (indicated by WUEi and δ^13^C) than WT plants both under control conditions and drought stress (Figure 5E,F). Previous reports revealed that cuticular waxes can enhance WUE in plants by reducing cuticular transpiration and water loss [8,9,49]. The Tr in MT leaves were much lower than those in WT leaves under control and drought conditions (Figure 5D). Therefore, we deduced that the significant increase in cuticular wax accumulation in MT leaves might enhance its WUE by reducing Tr and water loss under control and drought conditions. It is well-known that *L** represents the lightness or glossy degree in the color space system. Interestingly, a recent report suggested that blueberry leaves with high *L** could reduce heat load by reflecting a great amount of solar radiation, thus decreasing transpiration and consequently increasing WUE [50]. The cuticular waxes on a leaf surface reflect sunlight and consequently alter leaf lightness (*L**) [51]. Therefore, we deduced that the high level of *L** in MT leaves, which might be caused by increased cuticular wax accumulation, also contributed to its high WUE under control and drought conditions. Another explanation for the high WUE was the high Pn in MT leaves since WUE was reported to show higher dependence on Pn than Tr in citrus [52]. Taken together, we concluded that the increase in cuticular wax accumulation decreased cuticular water loss, increased *L** levels and consequently led to the increase in WUE in MT leaves. In addition, the increased level of Pn might be another reason for the increase in WUE in MT leaves.

### 3.3. The Changes in Expression Levels of Wax Biosynthesis and Export Genes Contributed to the Alterations in Cuticular Wax Accumulation of MT Leaves

To further explain the phenotype difference between WT and MT plants at the molecular level, transcriptome sequencing was performed on leaves of two varieties. KEGG enrichment analysis revealed that the ABC transporters and cutin and the suberine and wax biosynthesis were the fifth and sixth greatest enrichment pathways, suggesting the important contribution of wax biosynthesis and export DEGs to the phenotype difference between WT and MT plants (Figure 7D). Transcriptome sequencing identified three upregulated (*CsCER3-LIKE*, *CsABCG11-LIKE* and *CsABCG21-LIKE*) and five downregulated DEGs (*CsFAR1-LIKE*, *CsFAR3-LIKE*, *CsFAR4-LIKE*, *CsNSDHL-LIKE* and *CsChDI-LIKE*) involved in wax biosynthesis and transport from MT vs. WT (Table 1). The CER3 encodes a very-long-chain aldehyde reductase, which may physically interact with cytochrome b5 isoforms (CYTB5) and CER1 to catalyze the biosynthesis of n-alkanes [53,54]. Thus, the significant increase in the expression level of *CsCER3-LIKE* may explain the sharp increase in n-alkane amounts in MT leaves (Table 1 and Figure 10A). To date, numerous ABC transporter G subfamily genes, such as *AtABCG11* and *AtABCG12* from Arabidopsis, *OsABCG9* from rice, *PpABCG7* from *Physcomitrella patens* and *glossy13* (*AtABCG32* homolog) from maize, have been reported to export the cuticular waxes from plasma membrane (PM) to an apoplastic environment [55,56,57,58,59]. In this study, the expression levels of two ABC transporter G subfamily DEGs (*CsABCG11-LIKE* and *CsABCG21-LIKE*) in MT leaves were much higher than those in WT leaves under control and drought stress (Table 1 and Figure 10G,H). The increased expression of these two genes might improve the cuticular wax export from PM to the extracellular environment, leading to the increase in amounts of cuticular wax components in MT leaves.

The sterol-4alpha-carboxylate 3-dehydrogenase (NSDHL) and cholestenol delta-isomerase (ChDI) are involved in the biosynthesis of sterols [60]. The decline in the expression levels of *CsNSDHL-LIKE* and *CsChDI-LIKE* might cause a reduction in the amounts of sterols in MT leaves (Table 1 and Figure 10E,F).

In plants, fatty acyl-CoA reductase (FAR) catalyzes the reduction in VLC acyl-CoA to n-primary alcohols [61,62,63]. To our surprise, the expression levels of three *CsFAR-LIKE* genes (*CsFAR1-LIKE*, *CsFAR3-LIKE* and *CsFAR4-LIKE*) declined in MT leaves, which was contradictory to the increase in the amounts of n-primary alcohols (Table 1 and Figure 10B–D). This result suggested that other unidentified genes except for *CsFARs* might be involved in the biosynthesis of n-primary alcohols. Another explanation for this divergence was that the increases in the expression levels of *CsABCG11-LIKE* and *CsABCG21-LIKE* exported much more n-primary alcohols from PM to the extracellular environment, resulting in the increase in the amounts of n-primary alcohols in MT leaves. Interestingly, the expression levels of all wax-related genes in WT and MT leaves increased after drought stress, which suggested that these genes might be involved in navel orange response to drought stress (Figure 10A–H).

Previous studies revealed that numerous transcription factors (TFs), including the MYB family (MYB16, MYB30, MYB41, MYB94, MYB96, MYB106, MYB107 and MIXTA1), AP2/ERF family (SHN1, SHN2, SHN3, WXP1, WXP2, WRI4, DEWAX, RAP2.4), WRKY family (WRKY20), HD-ZIP IV family (OCL1 and Woolly), WW domain protein (CFL1) and MYC family (MYC2), can bind to the promoter of downstream wax biosynthesis genes to activate or repress their expression and influence plant tolerance to drought stress by regulating cuticular wax accumulation [5,64,65,66,67,68,69,70,71,72,73,74,75,76,77]. In this study, we identified 74 DEGs encoding TFs in MT vs. WT. Of these genes, 10 AP2/ERF family genes, 10 WRKY family genes and 9 MYB family genes were identified in all DEGs (Figure 8B and Appendix A). However, none of these genes was reported to be involved in cuticular wax biosynthesis. Further research should be performed to confirm whether these genes are related to cuticular wax biosynthesis in citrus.

Above all, we concluded that the changes in the expression levels of wax biosynthesis and export genes led to the alterations in the number of cuticular wax components and, finally, improved the drought tolerance and WUE in MT plants.

### 3.4. The DEGs Involved in the MAPK Signaling Pathway-Plant, ROS Scavenging and Other Enriched Pathways Might Contribute to Improve MT Tolerance to Drought Stress

The most significantly enriched pathway is the MAPK signaling pathway plant. It has been reported that the MAPK signaling pathway plays an important role in regulating the plant response to drought [78]. As the first member of the MAPK cascade, MAPKKK, also named MEKK, are activated by extracellular stimuli and then activates MAPKK and MAPK downstream by relay phosphorylation [79]. The MEKK family genes were reported to be involved in the plant’s response to drought stress [80,81,82,83]. In the present study, the expression of *CsMEKK1-LIKE* was upregulated in MT leaves under control and drought stress, suggesting this gene might play an important role in enhancing MT drought tolerance (Table 1, Figure 8A and Figure 10I). Interestingly, a rice *MEKK1* gene, *DSM1*, has been reported to enhance plant drought tolerance by promoting ROS scavenging [84]. Thus, the upregulation of *CsMEKK1-LIKE* expression might promote ROS scavenging and reduce H_2_O_2_ levels in MT leaves. In fact, most of the MAPK signaling pathway genes were upregulated in MT leaves and involved in signal transduction, environmental adaption and metabolism of stress-related chemical compounds, indicating the important role of the MAPK signaling pathway in enhancing MT resistance to drought stress (Figure 8A and Figure 9A, Appendix A).

In addition to the MAPK signaling pathway, many DEGs were also enriched in numerous pathways involved in the plant’s response to drought and other abiotic stresses, such as phenylpropanoid biosynthesis, flavonoid biosynthesis, brassinosteroid biosynthesis, glutathione metabolism, starch and sucrose metabolism, etc., suggesting the potential role of these DEGs in the citrus response to drought stress (Figure 7D and Appendix A). Interestingly, three DEGs (*CsSOD1-LIKE*, *CsPRX5-LIKE* and *CsPRX10-LIKE*) encoding antioxidant enzymes were upregulated in MT leaves under control and drought conditions (Table 1 and Figure 10J–L). Plant SOD and PRX family genes encode superoxide dismutase and peroxidase, respectively, which are involved in the ROS scavenging system [41]. Further study revealed that overexpression of *SOD* and *PRX* genes could enhance plant drought tolerance [85,86]. Thus, the upregulated expression of *CsSOD1-LIKE*, *CsPRX5-LIKE* and *CsPRX10-LIKE* might enhance the ROS scavenging capacity of MT plants to reduce H_2_O_2_ levels under drought stress (Figure 4J) and finally result in the increase in MT drought resistance. These results suggested that the DEGs involved in the MAPK signaling pathway plant, ROS scavenging and other enriched pathways might also contribute to improving MT drought tolerance.

### 3.5. The DEGs Encode Transcription Factor May Play an Important Role in Enhancing MT Tolerance to Drought Stress

The transcription factor can positively or negatively regulate numerous downstream gene expressions at the transcript level. Our study identified that 74 DEGs encode transcription factors (30 upregulated DEGs and 44 downregulated DEGs) belonged to 24 transcription factor families, such as AP2/ERF, WRKY, MYB, C2H2, NAC, MADS, FAR1, Dof, bHLH, G2-like and GRAS, etc. (Figure 8B and Appendix A). Notably, AP2/ERF and WRKY were the top two largest transcription factor families, containing 10 DEGs in each family. A large number of AP2/ERF family genes from sesame [87], cauliflower [88], tobacco [89] and strawberry [90] and WRKY family genes from citrus [91], common bean [92] and peanut [93] were reported to be involved in the response to drought stress. The remaining transcription factor families, such as NAC, MYB, bZIP, bHLH, C3H, MADS, GRAS and Dof, also played an important role in the drought stress response in plants [94]. In addition, 17 DEGs encoding transcription factors were involved in signal transduction, 15 DEGs were classified as environmental adaption and 7 DEGs were related to the metabolism of stress-related chemical compounds (Figure 9B and Appendix A). These results suggested that at least some of these transcription factors were related to the citrus response to drought stress. Previous studies showed that *ERF4* [95], *ERF9* [96], *MYB62* [97], *ZAT10* [98] and *NAC22* [99] could positively regulate plant tolerance to drought stress, whereas *WRKY27* [100] and *WRKY29* [101] were negatively related to plant drought response. In the present study, the upregulated expression of *CsERF4-LIKE*, *CsERF9-LIKE*, *CsMYB62-LIKE*, *CsZAT10-LIKE1*, *CsZAT10-LIKE2*, *CsNAC22-LIKE* and downregulated expression of *CsWRKY27-LIKE* and *CsWRKY29-LIKE* were observed in MT leaves both under control and drought conditions, suggesting these transcription factors might play important roles in enhancing MT tolerance to drought stress (Table 1, Figure 8B and Figure 10M–T).

## 4. Materials and Methods

### 4.1. Plant Materials

Seeds of trifoliate orange (*Poncirus trifoliata* (L.) Raf.) obtained from fruits on trifoliate orange trees in an orchard of Jiangxi Agriculture University (Nanchang, Jiangxi province, China) were sown in plastic pots filled with nutritional soil (garden soils, peats and sands mixed at the ratio of 3: 2: 1) to produce rootstocks. The fresh buds cut from the shoots of ‘Newhall’ navel orange (*Citrus sinensis* Osbeck cv. Newhall, WT) and ‘Longhuihong’ navel orange (*Citrus sinensis* Osbeck cv. Longhuihong, MT) were grafted on one-year-old trifoliate orange rootstocks to generate grafted plants of WT and MT. The WT and MT grafted plants were grown in a growth chamber under normal growth conditions (25 °C, relative humidity of 75%, 16 h light/8 h dark in a day). In this paper, a total of 200 one-year-old grafted plants per variety with uniform size were used as plant materials, of which 100 plants were used for control (under normal growth conditions) and 100 plants underwent drought treatment. The leaves used in all experiments were sampled from five plants randomly selected from 100 plants in each variety per biological replicate.

### 4.2. Analysis of Leaf Chromatic Aberration, Fresh and Dry Weight, Water Loss Rate and Chlorophyll Leaching Rate

In total, 15 fully expanded leaves per variety were used to determine leaf lightness and color by a colorimeter (CR-300, Minolta, Osaka, Japan). The colorimeter was calibrated with a standard white plate using an illuminant D65, 2° observer, Diffuse/O mode, 8 mm aperture. The colorimeter was standardized with a white tile. The *L** values represent lightness, with scores ranging from 0 (black) to 100 (white). The *a** values measure red–green colors, with positive scores indicating redness and negative scores representing greenness. The *b** values reflect yellow to blue colors, with positive values representing yellowness and negative values indicating blueness. The leaf fresh and dry weight were measured by an electronic balance (Model MP31001, Selon Scientific Instrument, Shanghai, China) and expressed as the average value of 20 leaves per replicate. Three biological replicates were used for each variety. The leaves were dried completely at 65 °C using a thermostatic oven (RX-RF023, kaimiao Electric equipment co., LTD, Suzhou, China).

The WT and MT plants were transferred to a dark environment for 6 h to make the stoma close. Then, the leaves were detached and placed in the dark (25 °C, relative humidity of 75%) for dehydration. As a result, 15 leaves per biological replicate in each variety were weighted at 1, 2, 4, 8, 12, 24, 36 and 48 h after detachment by an electronic balance (Model MP31001, Selon Scientific Instrument, Shanghai, China). The water loss rate was expressed by the percentage of the lost weight in initial leaf weight.

In total, five fully expanded leaves with the same size and weight per biological replicate in each variety were sampled from WT and MT plants. These leaves were soaked in 80% ethanol and kept in a dark environment (25 °C, relative humidity of 75%). After 2, 5, 8, 10, 24, 30h and 48 h of initial immersion, 2 mL of the solution was used to determine the absorbance of the extract at 647 nm (A647) and 664 nm (A664) wavelengths by a spectrophotometer (UV-2600, Shimadzu, Kyoto, Japan). The leaf chlorophyll concentration was calculated as the following equation: total micromoles chlorophyll = 7.93 × A664 + 19.53 × A647. The chlorophyll leaching rate was expressed as a percentage of the chlorophyll concentration at each time point in total chlorophyll extracted after 48 h. Three biological replicates were used for water loss and chlorophyll leaching experiments.

### 4.3. SEM Analysis

The SEM analysis was carried out according to our previous report [22]. In short, leaf disks with a diameter of 0.5 cm were excised from the fully expanded leaves and transferred to the aluminum holders. Then, the disks were frozen in liquid N2, freeze-dried and sputter-coated with gold film in a sputter coater (SBC-12, KYKY, Beijing, China). The coated disks were observed by SEM (Zeiss DSM 962, Carl Zeiss, Oberkochen, Germany) at ×1000 and ×3000 magnification.

### 4.4. Cuticular Wax Extraction and Analysis

The fully expanded leaves of WT and MT plants were collected for cuticular wax analysis. The surface areas of leaves were determined by counting pixels of leaf photos using Image J software. In total, 10 leaves from each variety per biological replicate were immersed three times in chloroform for 30 s each time to extract the cuticular waxes. Then, 5 μg n-tetracosane was added to the cuticular wax solution as the internal standard. The extracted solution was concentrated using a rotary evaporator under reduced pressure at 35 °C and dried under nitrogen. Before the gas chromatography-mass spectrometer (GC-MS) analysis, the extracts were derivatized with bis-N,O-(trimethylsilyl) trifluoroacetamide (BSTFA) in pyridine for 40 min at 70 °C to convert the free carboxyl and hydroxyl compounds to their corresponding trimethylsilyl (TMSi) ethers and esters. The derivatized extracts were then treated with a nitrogen stream to remove the residual BSTFA and re-dissolved in 1 mL of chloroform.

The GC-MS analysis was performed according to our previous report [26]. In brief, 1 μL of cuticular wax extract from each sample was used to identify the wax components by a GC system (Agilent 6890N, Agilent Technologies, Santa Clara, CA, USA) equipped with an HP-5 MS capillary column (30 m × 0.25 mm i.d. × 0.25 μm, Agilent Technologies, Santa Clara, CA, USA) and coupled with an MS detector (Agilent 5973N, Agilent Technologies, Santa Clara, CA, USA). The carrier gas was helium at a constant flow rate of 2 mL min^−1^. The on-column injection temperature of GC was 50 °C, 1 min, then increased to 170 °C by 20 °C min^−1^, remained at 170 °C for 2 min, increased to 300 °C by 5 °C min^−1^, remained at 300 °C for 8 min. The identification of cuticular wax compounds was performed by matching their mass spectra with those from the NIST 14 MS library. The quantification of cuticular wax components were performed by the same GC detector equipped with a flame ionization detector (FID) under the same chromatographic conditions. The wax compounds were quantified by comparing their peak areas with the internal standard. The amounts of cuticular wax components were expressed as microgram per leaf area (μg cm^−2^). Three biological replicates were used for each variety.

### 4.5. Analysis of Physiological Indexes in WT and MT Leaves under Control and Drought Treatment

For drought treatment, one-year-old WT and MT plants were cultivated without watering for 21 days and then used for further study. In contrast, the control plants were well watered in the same growth chamber. In each variety, 15 fully expanded leaves were used to measure the photosynthesis parameters at 10:00 a.m. The photosynthesis parameters of WT and MT leaves, including PnGs, Ci and Tr, were investigated by a LI-6400 photosynthetic system (LI-COR, Inc. Lincoln, NE, USA) according to the manufacturer’s instruction under a photon fux density of 1000 μmol m^−2^ s^−1^. The ambient carbon dioxide concentration (375 μmol mol^−1^) was controlled by the LI-6400 CO_2_ injecting system. Pn, Gs, Ci and Tr were automatically recorded by the photosynthetic system. The Pn/Tr was used to calculate WUEi.

In total, 15 fully expanded leaves per biological replicate in each variety were sampled and pooled for measurement of the physiological indexes. The ion leakage was measured by the method of a previous study [102]. In brief, the leaves were stripped and transferred to 30 mL of distilled water and then shaken by a gyratory shaker (200 rpm) at room temperature for 2 h. The initial conductivity (C1) was detected by a DDSJ-318 conductivity meter (Yidian Scientific Instrument Co., Ltd., Shanghai, China). Afterward, the samples were boiled for 10 min to reach maximum ion leakage. After cooling down at room temperature, the electrolyte conductivity (C2) was determined by a conductivity meter. Finally, the ion leakage (%) was calculated by 100× C1/C2. The contents of malondialdehyde (MDA), proline, soluble sugar, chlorophyll, H_2_O_2_ and the activities of SOD, POD, CAT were measured by corresponding detection kits (Nanjing Jiancheng Bioengineering Institute, Nanjing, China) according to the manufacturer’s protocols. Briefly, the MDA contents were measured by a thiobarbituric acid (TBA)-based colorimetric method. MDA was extracted from 0.1 g leaves homogenized in 1 m of 80% ethanol solution on ice. After centrifuging at 10,000× *g* for 20 min at 4 °C, the supernatants were extracted and mixed with 0.5 mL of 20% trichloroacetic acid containing 0.65% TBA. Then, the mixture was incubated at 95 °C for 30 min and cooled in an ice bath. Afterward, the mixture was centrifugated at 10,000× *g* for 10 min. The absorbance of the extracted supernatants was detected at 532 nm (A532), subtracting the value for nonspecific absorption at 600 nm (A600). The MDA content was calculated by the following equation: the MDA content = [6.45 × (A532 − A600)]/0.1. The ninhydrin reaction method was used to measure the proline content. In brief, a standard curve was obtained from a series of standard solutions containing 0–10 μg proline. A total of 0.5 g leaves was homogenized in 5 mL of 3% suphosalicylic acid and heated at 100 °C for 10 min. Then, 2 mL of the extracted solution was mixed with 2 mL of acetic acid and 2 mL of 2.5% acid ninhydrin reagent. This mixed solution was heated at 100 °C for 30 min and then cooled at room temperature. Afterward, 4 mL of methylbenzene was added to the solution and incubated for 10 min. The supernatants were isolated and centrifugated at 10,000× *g* for 5 min. The methylbenzene solution was used as a control, and the absorbance of the supernatants (2 mL) was detected at 520 nm (A520). The proline content was determined by the following equation: the proline content = (B × 5)/(0.5 × 2). B represents the proline content obtained by the standard curve according the A520. The phenol reaction method was used to detect the soluble sugar content. Briefly, a standard curve was obtained from a series of standard solutions containing 0–100 μg sucrose. A total of 0.2 g of leaves was boiled in 5 mL of distilled water for 30 min and diluted with distilled water to 10 mL. Then, 2 mL of the solution was added to 1 mL of 9% phenol and 5 mL of concentrated sulfuric acid. After standing for 30 min, distilled water was used as a control to determine the absorbance of the aqueous solution (2 mL) at 485 nm. The soluble sugar content was determined by the following equation: the soluble sugar content = (D × 10)/(0.2 × 2). D represents the soluble sugar content obtained by the standard curve according to the A485. To measure the chlorophyll content, 0.1 g of fine powder of leaves was homogenized in 1 mL of 80% acetone and held for 15 min at room temperature in the dark. The extracted solution was centrifugated at 10,000× *g* for 20 min. Then, the supernatant was used to measure the absorbance at 663 and 645 nm. The chlorophyll content was calculated by the following equation: the chlorophyll content = (20.29 × A645 + 8.05 × A663). For H_2_O_2_ analysis, 0.5 g of fine powder of leaves was homogenized in 2 mL cold acetone. The homogenized solutions were centrifuged at 10,000× *g* for 10 min. Then, the supernatants were mixed with titanium reagent (0.2 mL of 20% titanic tetrachloride in concentrated HCl) and 0.4 mL of NH_4_OH. Then, the mixtures were centrifuged at 12,000× *g* for 5 min. The precipitates were solubilized in 2 mL 2 N H_2_SO_4_, washed repeatedly with acetone and brought to a final volume of 2 mL. The absorbance of the final solution was measured at 415 nm against a water blank, which had been carried through the same procedure. The H_2_O_2_ content was calculated by comparing the absorbance against the standard curve representing the titanium-H_2_O_2_ complex in the range from 0.05 to 0.3 μmol·mL^−1^. For the extraction of SOD, POD and CAT, 0.5 g of leaf powder was homogenized in 5 mL of extraction buffer containing 50 mM phosphate buffer (pH7.8) and 1% polyvinylpyrrolidone. After centrifuging at 10,000× *g* for 20 min (4 °C), the supernatants were collected for enzyme activity analysis. The SOD reaction mixture contained 100 μL enzyme extract, 50 mM sodium phophate buffer (pH 7.8), 10 μM EDTA, 75 μM nitroblue tetrazolium (NBT), 13 mM methionine and 2 μM riboflavin in a total volume of 3 mL. The mixtures were exposed to white fluorescent illumination for 20 min, and the absorbance was measured at 560 nm. The SOD activity was expressed as U·g^−1^. One unit of SOD activity was defined as the amount of enzyme inhibiting NBT reduction by 50%. The POD reaction mixture contained 0.05 M phosphate buffer (pH 7.0), 0.3% H_2_O_2_, 0.2% guaiacol and 50 μL enzyme extract in a total volume of 3 mL. The POD activity, expressed as U·g^−1^, was determined based on the increase in absorbance read at 470 nm, and one unit of POD activity was defined as the increase in absorbance by 0.01 per min. The CAT reaction mixture was composed of 0.1% H_2_O_2_, 0.1 M phosphate buffer (pH 7.0) and 100 μL enzyme extract in a total volume of 3 mL. The CAT activity, expressed as U·g^−1^, was assessed by monitoring the decrease in absorbance at 240 nm as a consequence of H_2_O_2_ consumption, and one unit of CAT activity was defined as a reduction in the absorbance by 0.01 per min. The leaf stable carbon isotope composition (=δ^13^C) analysis was carried out according to a previous report [103]. Briefly, the leaf samples were dried in an oven at 70 °C and ground to a fine powder. Subsamples of the leaf powder were combusted in an isotope ratio mass spectrometer (IRMS) (DELTA V Plus, Thermo Fisher Scientific, Bremen, Germany). The resulting CO_2_ was separated and the ratio of ^13^C/^12^C was assessed by the IRMS. The δ^13^C (‰) was expressed relative to the Pee Dee Belemnite (PDB) standard and was calculated using the following equation: δ^13^C = [(Rsa − Rsd)/Rsd] × 1000. The Rsa and Rsd in the equation are the ratio of ^13^C/^12^C of the sample and the standard, respectively. Three biological replicates were used for the above experiments.

### 4.6. Total RNA Extraction and Transcriptome Sequencing

Three biological replicates were used for transcriptome sequencing. In total, 15 fully expanded leaves per biological replicate in each variety were sampled from WT and MT plants under normal growth conditions. These leaves were pooled and ground to a fine powder for total RNA extraction. Total RNA was extracted from WT and MT leaves according to our previous report [17]. The quantity, concentration and RNA Integrity Number (RIN) of the total RNA were assessed by an Agilent 2100 Bioanalyzer (Agilent, SantaClara, CA, USA). The poly(A) mRNA was obtained from total RNA using oligo(dT)-attached magnetic beads. The mRNA was interrupted to small fragments and then used to synthesize the first strand of cDNA using random hexamer primers. A Super Script Double-Stranded cDNA Synthesis kit (Invitrogen, Camarillo, CA, USA) was used to synthesize the second strand cDNA. After end repair and 5′ phosphorylation, the cDNA samples were 3′-adenylated and ligated to 3′ and 5′ adapters. After PCR amplification, the PCR products were separated into a single cDNA strand by heat denaturation. Circular single-stranded DNA libraries were obtained by a bridge primer and were sequenced on the DNBSEQ platform (BGI, Shenzhen, China).

The empty reads, adaptor sequences, reads with more than 5% unknown nucleotides, and low-quality sequences were removed from the raw data to obtain clear reads using SOAPnuke version 1.4.0 (https://github.com/BGI-flexlab/SOAPnuke) accessed on 28 July 2020. For gene annotation, all clean reads were mapped to the Citrus sinensis genome and gene (http://citrus.hzau.edu.cn/) using HISAT2 version 2.1.0 (http://www.ccb.jhu.edu/software/hisat) accessed on 28 July 2020 and Bowtie2 (http://bowtie-bio.sourceforge.net/Bowtie2/index.shtml) accessed on 28 July 2020, respectively. For transcription factor annotation, the open reading frames (ORFs) of genes were detected by GetORF (http://emboss.sourceforge.net/apps/cvs/emboss/apps/getorf.html) accessed on 28 July 2020. Then, these ORFs were aligned PlantTFDB (http://plntfdb.bio.uni-potsdam.de/v3.0/) using Hmmsearch vervison 3.0 (http://hmmer.org) accessed on 28 July 2020. Gene transcript levels were quantified by Expectation Maximization (RSEM) version 1.2.8 (http://deweylab.biostat.wisc.edu/rsem) accessed on 28 July 2020. Fragments per kilo base of transcript per million mapped reads (FPKM) were used to normalize the gene expression levels. In this study, only DEGs with |log_2_ (MT/WT)| ≥ 1 and Q-value (adjusted *p*-value) ≤ 0.001 were used for further study. The volcano plot was constructed by ggplot2 function in R software version 3.1.1 (Vienna University of Economics and Business, Vienna, Austria) accessed on 28 July 2020. The heatmaps were constructed by Tbtools version 1.098696 (South China Agricultural University, Guangzhou, Guanzhou, China) accessed on 28 July 2020. GO analysis and KEGG pathway analysis were carried out by phyper function in R software version 3.1.1 (accessed on 28 July 2020) with a threshold of Q-value ≤ 0.05 for significance. The transcriptome sequencing data have been submitted to the Sequence Read Archive (SAR) database under accession number PRJNA799891.

### 4.7. qRT-PCR Analysis

The same plant materials and drought treatment procedure as described in the physiological index investigation were used for qRT-PCR. The transcript levels of 16 wax biosynthesis, export and drought responsive DEGs in WT and MT leaves under control and drought conditions were investigated by qRT-PCR. The total RNA extraction and cDNA synthesis were performed as described in our previous study [22]. A series of gene specific primers were designed by Primer Premier 5.0 (Appendix A). The qRT-PCR was carried out according to our previous report [22].

### 4.8. Statistical Analysis

All data are shown as the means ± standard deviations. Student’s *t*-test in SPSS version 22 was used to compare the data differences between WT and MT. Statistical significance was considered at the *p* < 0.01 and *p* < 0.05 level.

## 5. Conclusions

In conclusion, the amounts of total waxes and aliphatic wax compounds, including n-alkanes, n-primary alcohols and n-aldehydes in MT leaves, were significantly higher than those in WT leaves. The increase in aliphatic wax accumulation reduced the cuticular permeability and, finally, improved the drought tolerance and WUE in MT plants. The contents of MDA and H_2_O_2_ in MT leaves were much lower than those in WT leaves under control and drought conditions. In contrast, MT leaves possessed much higher levels of proline and soluble sugar and significantly enhanced SOD, CAT and POD activities under control and drought conditions. These results suggested that the MT leaves had improved the capacity of ROS scavenging, suffered much less ROS damage and kept much better cell membrane stability, which might be another reason for its enhanced drought tolerance. Based on transcriptome sequencing and qRT-PCR, we concluded that several structural genes potentially involved in the wax biosynthesis and transport (*CsCER3-LIKE*, *CsABCG11-LIKE* and *CsABCG21-LIKE*), MAPK cascade (*CsMEKK1-LIKE*), ROS scavenging (*CsSOD1-LIKE*, *CsPRX5-LIKE* and *CsPRX10-LIKE*) and genes encoding transcription factors (*CsERF4-LIKE*, *CsERF9-LIKE*, *CsMYB62-LIKE*, *CsZAT10-LIKE1*, *CsZAT10-LIKE2*, *CsWRKY27-LIKE* and *CsWRKY29-LIKE*) might play an important role in increasing MT drought tolerance and WUE.

## Figures and Tables

**Figure 1 ijms-23-05660-f001:**
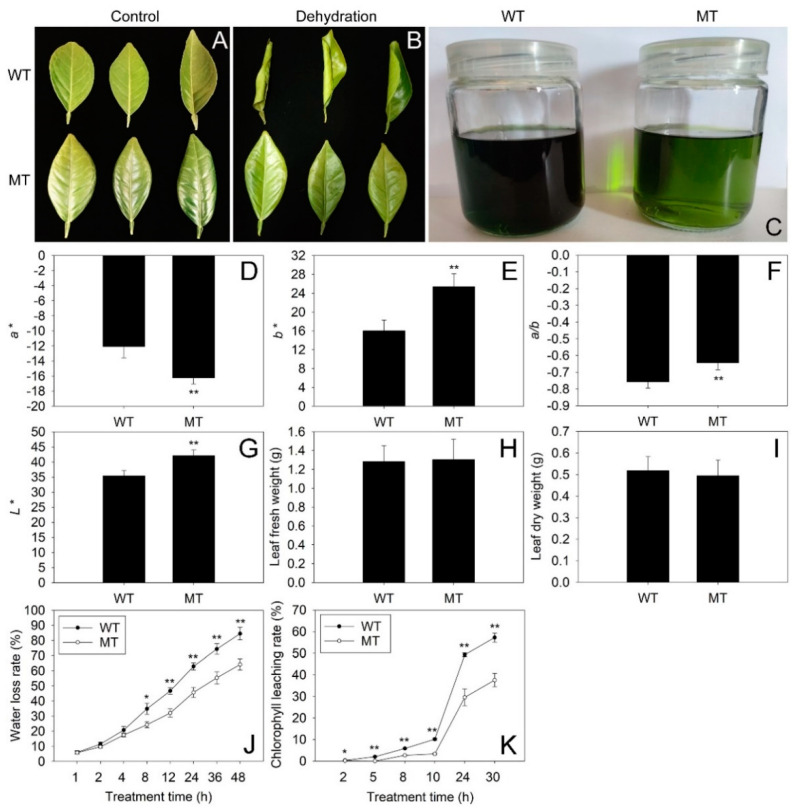
Analyses of phenotype, chromatic aberration and cuticular permeability in WT and MT leaves. (**A**,**B**) The phenotype of WT and MT leaves under control (well−watered) and dehydration stress (leaves detached and placed in dark condition for 48 h). (**C**) The chlorophyll leaching conditions of WT and MT leaves immersed in an alcohol solution for 30 h. (**D**–**G**) The values of *a**, *b**, *a**/*b** and *L** detected by a colorimeter. Vertical bars represent standard deviations of the means (*n* = 15). (**H**,**I**) The fresh and dry weight in WT and MT leaves. (**J**,**K**) The comparison of water loss rates and chlorophyll leaching rates in WT and MT leaves. Vertical bars represent standard deviations of the means (*n* = 3). Significant differences between WT and MT leaves at the *p* < 0.05 and *p* < 0.01 levels were indicated by * and **, respectively, according to Student’s *t*-test.

**Figure 2 ijms-23-05660-f002:**
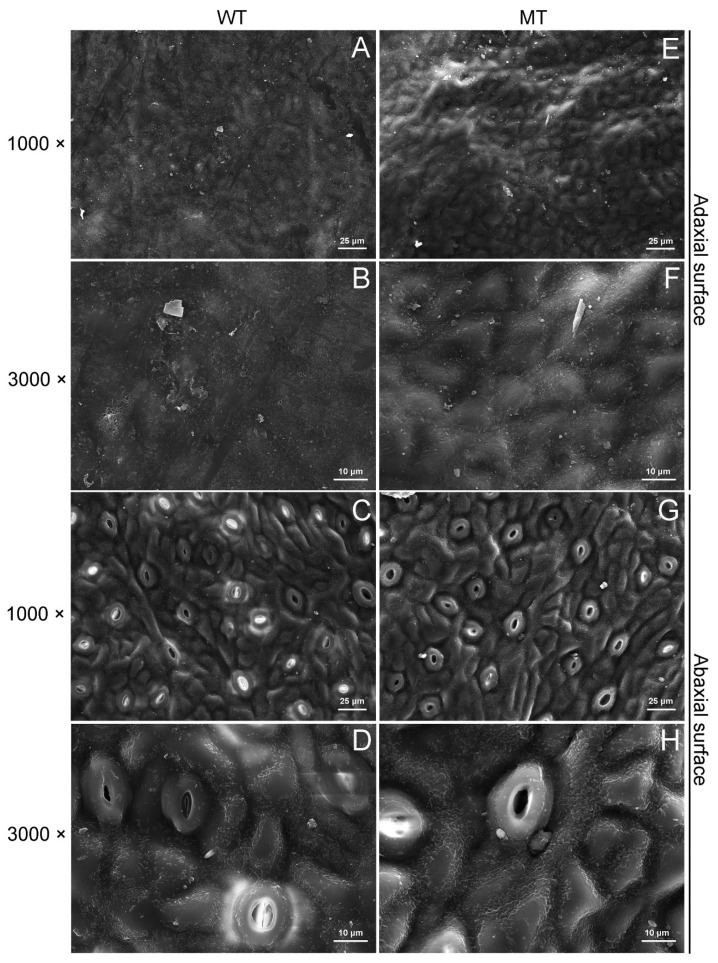
Scanning electron micrographs of cuticular waxes on the surfaces of WT and MT leaves. (**A**,**B**) The adaxial surfaces of WT leaves were imaged at 1000× and 3000× magnification. (**C**,**D**) The abaxial surfaces of WT leaves were imaged at 1000× and 3000× magnification. (**E**,**F**) The adaxial surfaces of MT leaves were imaged at 1000× and 3000× magnification. (**G**,**H**) The abaxial surfaces of MT leaves were imaged at 1000× and 3000× magnification.

**Figure 3 ijms-23-05660-f003:**
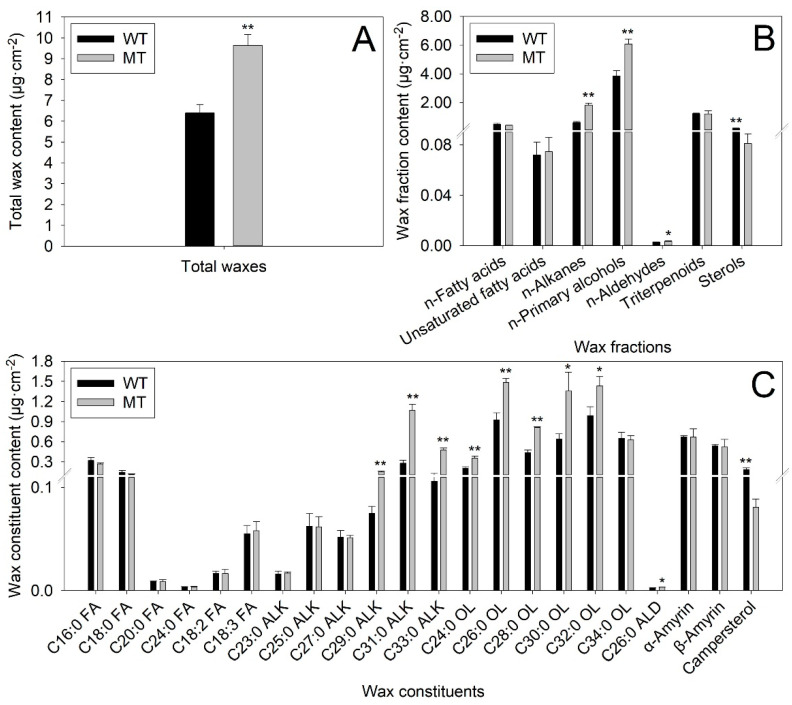
The contents of total waxes (**A**), wax fractions (**B**) and wax constituents (**C**) in WT and MT leaves. Abbreviations of the wax fractions are listed as follows: FA, fatty acid; ALK: alkane; OL: alcohol; ALD: aldehyde. Vertical bars represent standard deviations of the means (*n* = 3). Significant differences between WT and MT leaves at the *p* < 0.05 and *p* < 0.01 levels were indicated by * and **, respectively, according to Student’s *t*-test.

**Figure 4 ijms-23-05660-f004:**
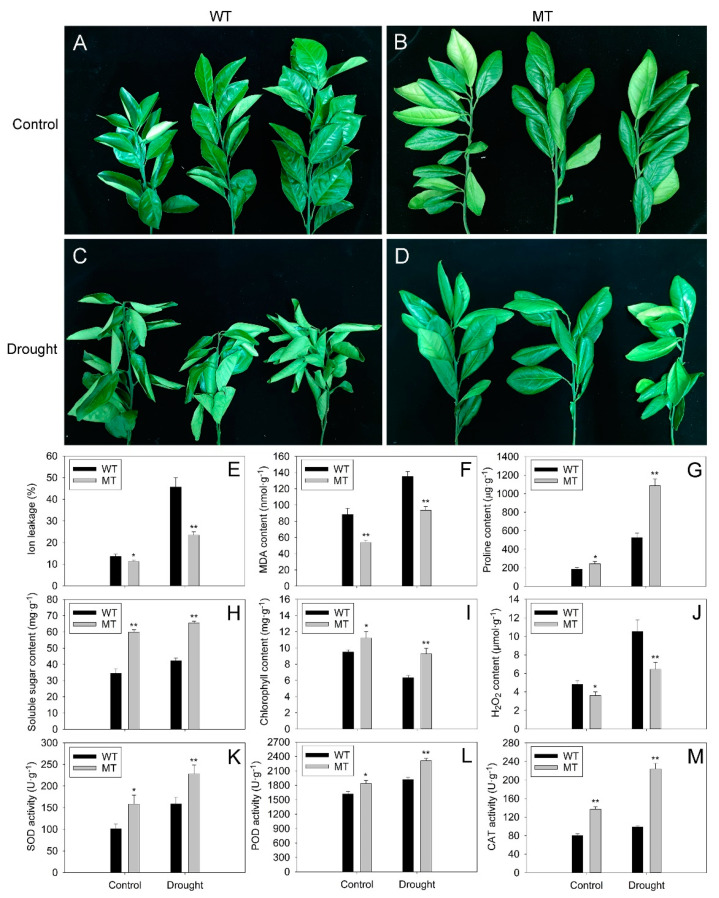
The morphological and physiological analyses of WT and MT plants under control (well-watered) and drought conditions (without watering for 21 days). (**A**–**D**) Phenotype of WT and MT seedings under control and drought conditions. The leaves of WT and MT leaves were collected to measure the (**E**) ion leakage, the contents of (**F**) MDA, (**G**) proline, (**H**) soluble sugar, (**I**) chlorophyll, (**J**) H_2_O_2_ and the activities of (**K**) SOD, (**L**) POD, (**M**) CAT. Vertical bars represent standard deviations of the means (*n* = 3). Significant differences between WT and MT leaves at the *p* < 0.05 and *p* < 0.01 levels were indicated by * and **, respectively, according to Student’s *t*-test.

**Figure 5 ijms-23-05660-f005:**
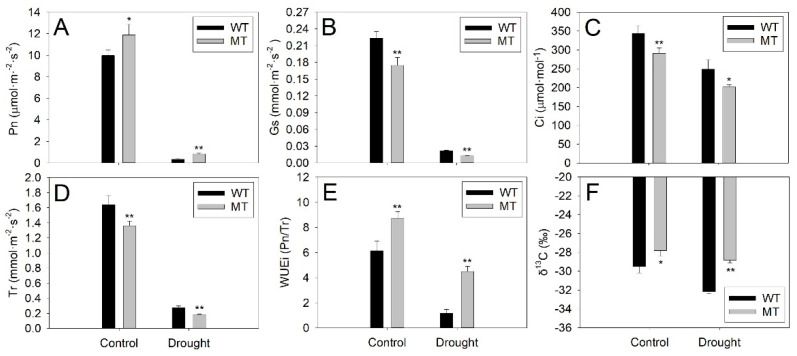
Analyses of photosynthetic indexes and water use efficiency (WUE) of WT and MT plants under control (well-watered) and drought conditions (without watering for 21 days). The (**A**) net photosynthetic rate (Pn), (**B**) stomatal conductance (Gs), (**C**) intercellular CO_2_ concentration (Ci) and (**D**) transpiration rate (Tr) were measured by a LI-6400 photosynthetic system. (**E**) The Pn/Tr was used to calculate instantaneous water use efficiency (WUEi). Vertical bars represent standard deviations of the means (*n* = 15). (**F**) The leaf’s δ^13^C were used to represent the short-term WUE in the whole plant. Vertical bars represent standard deviations of the means (*n* = 3). Significant differences between WT and MT leaves at the *p* < 0.05 and *p* < 0.01 levels were indicated by * and **, respectively, according to Student’s *t*-test.

**Figure 6 ijms-23-05660-f006:**
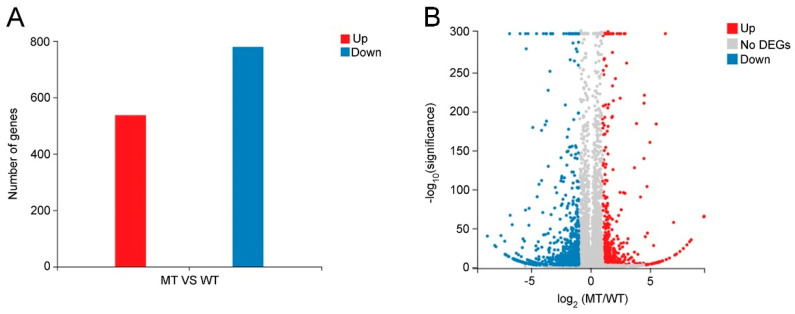
The differentially expressed genes (DEGs) identified in MT vs. WT according to the criteria of |log_2_ (MT/WT)| ≥ 1 and Q-value < 0.001. (**A**) The number of upregulated and downregulated DEGs. (**B**) The volcano plot of DEGs. The red and blue colors indicate upregulated and downregulated DEGs, respectively. The gray color indicates genes with no significant expression difference between WT and MT.

**Figure 7 ijms-23-05660-f007:**
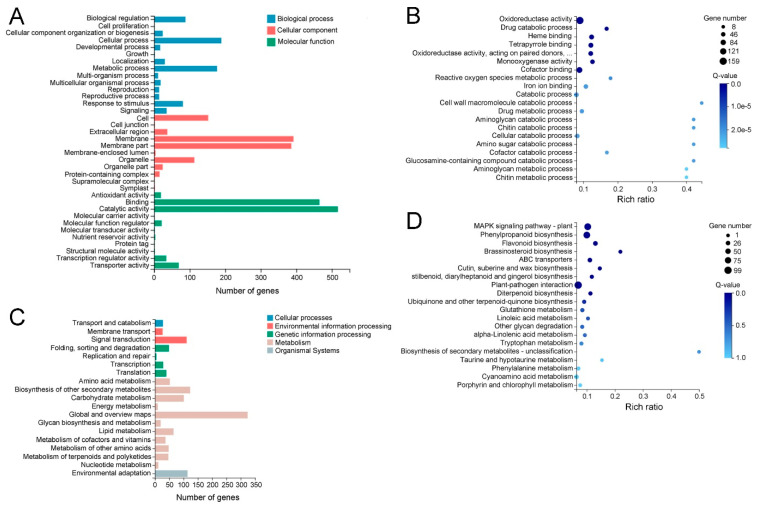
GO and KEGG analyses of differentially expressed genes (DEGs) in MT vs. WT. (**A**,**B**) GO classification and enrichment analyses. (**C**,**D**)The classification and enrichment analyses of KEGG pathways of DEGs in MT vs. WT. The rich ratio is the ratio of the number of DEGs annotated in a given GO term or pathway to the number of all genes annotated in the GO term or pathway.

**Figure 8 ijms-23-05660-f008:**
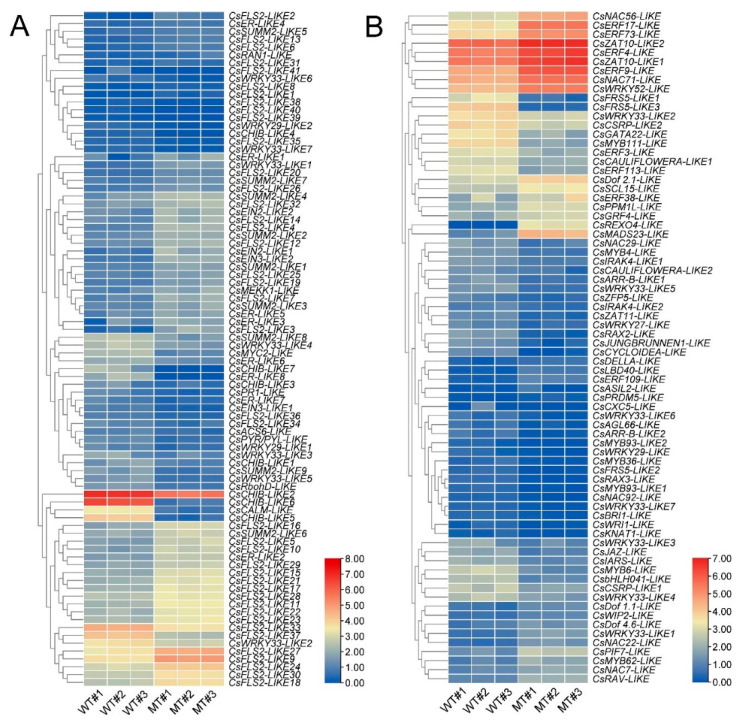
The heat maps show expression levels of differentially expressed genes (DEGs) involved in (**A**) the MAPK signaling pathway plant and (**B**) DEGs encoding transcription factors in WT and MT samples. The gradient color barcode indicates log_2_(FPKM+1) values in each sample.

**Figure 9 ijms-23-05660-f009:**
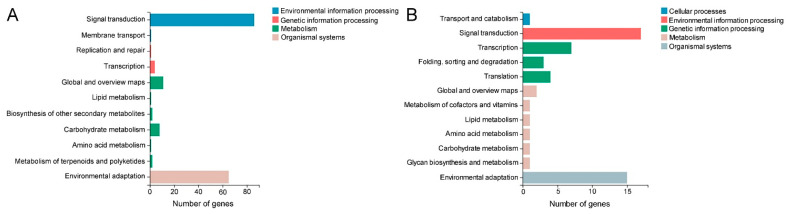
KEGG classification analyses of differentially expressed genes (DEGs) in (**A**) MAPK signaling pathway plant and (**B**) DEGs encoding transcription factors.

**Figure 10 ijms-23-05660-f010:**
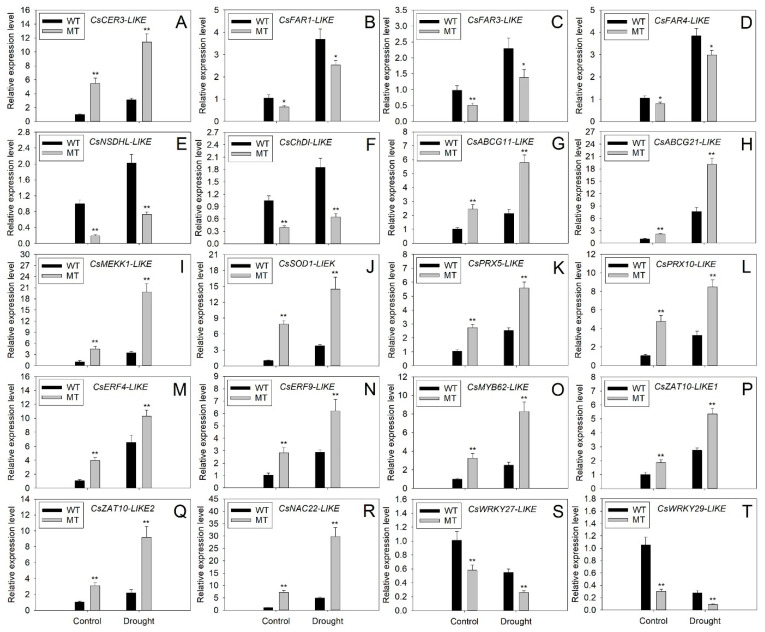
Expression analysis of structural differentially expressed genes (DEGs) involved in (**A**–**F**) wax biosynthesis and (**G**,**H**) export, (**I**) MAPK cascade, (**J**–**L**) ROS scavenging and (**M**–**T**) DEGs encoding transcription factors under control (well-watered) and drought conditions (without watering for 21 days). The y-axis records the relative gene expression levels calculated by the 2^−^^ΔΔ^^CT^ method with citrus *β*-*actin* as the endogenous reference. Vertical bars represent standard deviations of the means (*n* = 3). Significant differences between WT and MT leaves at the *p* < 0.05 and *p* < 0.01 levels were indicated by * and **, respectively, according to Student’s *t*-test.

**Table 1 ijms-23-05660-t001:** The differentially expressed genes involved in wax biosynthesis and transport, MAPK cascade, ROS scavenging and encoded drought-responsive transcription factors.

Gene Name	Gene ID	log_2_ (MT/WT)	Function Definition
*CsCER3-LIKE*	Cs4g02580	2.8143	Very-long-chain aldehyde decarbonylase
*CsFAR1-LIKE*	Cs8g15290	−2.3009	Fatty acyl-CoA reductase 1
*CsFAR3-LIKE*	Cs5g15350	−1.3315	Fatty acyl-CoA reductase 3
*CsFAR4-LIKE*	Cs5g15345	−2.0485	Fatty acyl-CoA reductase 3
*CsNSDHL-LIKE*	BGI_novel_G000534	−4.7498	Sterol-4alpha-carboxylate 3-dehydrogenase
*CsChDI-LIKE*	orange1.1t04155	−1.1913	Cholestenol Delta-isomerase
*CsABCG11-LIKE*	orange1.1t01993	1.4780	ATP-binding cassette, subfamily G, member 11
*CsABCG21-LIKE*	Cs7g12620	1.2926	ATP-binding cassette, subfamily G, member 21
*CsMEKK1-LIKE*	Cs5g22020	1.2434	MAP kinase kinase kinase
*CsSOD1-LIKE*	orange1.1t05755	2.2530	Superoxide dismutase 1, Cu-Zn family
*CsPRX5-LIKE*	orange1.1t01747	1.8422	Peroxidase 5
*CsPRX10-LIKE*	Cs5g34200	2.8727	Peroxidase 10
*CsPRX24-LIKE*	Cs9g05140	1.1134	Peroxidase 24
*CsPRX25-LIKE*	Cs4g17860	1.4794	Peroxidase 25
*CsERF4-LIKE*	Cs1g07950	1.0323	Ethylene-responsive transcription factor 4
*CsERF9-LIKE*	Cs2g23660	1.5729	Ethylene-responsive transcription factor 9
*CsMYB62-LIKE*	Cs7g26930	1.3835	Transcription factor MYB62
*CsZAT10-LIKE1*	Cs3g15900	1.1106	Zinc finger protein ZAT10
*CsZAT10-LIKE2*	Cs8g04280	1.0356	Zinc finger protein ZAT10
*CsNAC22-LIKE*	Cs5g29650	2.0431	NAC domain-containing protein 22
*CsWRKY27-LIKE*	Cs9g19070	−1.5064	WRKY transcription factor 27
*CsWRKY29-LIKE*	Cs5g03010	−4.1799	WRKY transcription factor 29

## Data Availability

Transcriptome data are available at NCBI SRA accession PRJNA799891.

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
