# Peer review of "Transcriptome and Physiological Analyses of a Navel Orange Mutant with Improved Drought Tolerance and Water Use Efficiency Caused by Increases of Cuticular Wax Accumulation and ROS Scavenging Capacity"

_ijms, 2022, doi:10.3390/ijms23105660_

Round 1
Reviewer 1 Report
In this work, by means of transcriptomic approach and real-time quantitative PCR, the differential expression of the main pathways involved in the production and accumulation of waxes in the leaves of MT Longhuihong species, compared to WT species, has been highlighted. The work is quite well written and structured, however there are several critical issues to be resolved, below some specific comments:
Line 41: add references
Line 56-63: rearrange to make the text easier to read.
Line 82: real-time quantitative PCR (qRT-PCR)
Line 608: are they certified seeds? Specify their origin
Line 609: specify what you mean by nutritional soil
Line 617: do the leaves used come from the same plant or from a randomized sampling? How many plants did you use? This information that is given later in the text should be specified earlier.
Line 88: " observed by naked eyes" does not seem like a very scientific parameter, consider removing this statement from the text.
Figure 2: the images are of low quality and particularly blurred, especially A E B F, is it not possible to replace them?
Figure 4 and 5: How ion leakage, the contents of MDA, proline, soluble sugar, chlorophyll, H2O2, the activities of SOD, POD and CAT, stomatal conductance, intercellular CO2 concentration and transpiration rate were measured should be indicated in Materials and Methods.
Line 197: in materials and methods it is stated that RNA extraction was performed on a composite sample consisting of 15 leaves, here instead is indicated a number of 6 leaves per biological replicate... so?
Line 197: was only the total mRNA sequenced? Or was all the RNA sequenced and then a filter was applied to remove the rRNA?
Author Response
Dear reviewer:
Thank you very much for your comment. We revised the manuscript in accordance with the reviewers’ comments, and carefully proof-read the manuscript to minimize typographical, grammatical, and bibliographical errors. Here below is our description on revision according to the reviewers’ comments.
Point-by-point answers to reviewer #1:
In this work, by means of transcriptomic approach and real-time quantitative PCR, the differential expression of the main pathways involved in the production and accumulation of waxes in the leaves of MT Longhuihong species, compared to WT species, has been highlighted. The work is quite well written and structured, however there are several critical issues to be resolved, below some specific comments:
Line 41: add references
The authors’ Answer: According to the reviewer’s comment, we added a reference [2] in the revised manuscript (line 42).
Line 56-63: rearrange to make the text easier to read.
The authors’ Answer: According to the reviewer’s comment, we rewritten this part in line 57-74 of the revised manuscript.
Line 82: real-time quantitative PCR (qRT-PCR)
The authors’ Answer: According to the reviewer’s comment, we used ‘real-time quantitative PCR (qRT-PCR)’ to replace ‘real-time quantitative RT-PCR’ in line 100 of the revised manuscript.
Line 608: are they certified seeds? Specify their origin
The authors’ Answer: Thank you for the reviewer’s comment. As stated in ‘Plant materials’, the WT and MT plants used in this study were grafted seedlings of navel orange (WT and MT buds grafted on rootstocks of trifoliate orange). Trifoliate orange is a common citrus rootstock in China. Before grafting, one-year-old rootstock seedlings of trifoliate orange must be produced by sowing seeds into the soil. The seeds of trifoliate orange fruits were obtained from trifoliate orange trees in an orchard of Jiangxi agriculture university. Thus, we certified the seeds used here is correct.
In addition, we added the seeds origin in line 692-694 of the revised manuscript.
Line 609: specify what you mean by nutritional soil
The authors’ Answer: Thank you for the reviewer’s comment. The nutritional soil is composed of garden soils, peats and sands at the ratio of 3: 2: 1. We have stated the components of nutritional soil in the revised manuscript (line 694-695).
Line 617: do the leaves used come from the same plant or from a randomized sampling? How many plants did you use? This information that is given later in the text should be specified earlier.
The authors’ Answer: Thank you for the reviewer’s comment. In this paper, a total of 200 one-year-old grafted plants per variety with uniform size were used as plant materials for all experiments, of which 100 plants were used for control (under normal growth conditions) and 100 plants underwent drought treatment. The leaves used in all experiments were sampled from five plants randomly selected from 100 plants in each variety per biological replicate. We put this statement in the end of ‘4.1. Plant materials’ (line 700-705) in the revised manuscript
Line 88: "observed by naked eyes" does not seem like a very scientific parameter, consider removing this statement from the text.
The authors’ Answer: According to the reviewer’s comment, we deleted this statement from the text in the revised manuscript (line 106-107).
Figure 2: the images are of low quality and particularly blurred, especially A E B F, is it not possible to replace them?
The authors’ Answer: Thank you for the reviewer’s comment. The original picture is clearer than that paste in the manuscript. Maybe pasting to the Microsoft word reduce the figure quality. We also increased the resolution from 600 dpi to 1000 dpi to improve the figure quality.
Figure 4 and 5: How ion leakage, the contents of MDA, proline, soluble sugar, chlorophyll, H2O2, the activities of SOD, POD and CAT, stomatal conductance, intercellular CO2 concentration and transpiration rate were measured should be indicated in Materials and Methods.
The authors’ Answer: According to the reviewer’s comment, we added the methods of these physiological indexes in the revised manuscript (line 778-781, 785-791, 794-852).
Line 197: in materials and methods it is stated that RNA extraction was performed on a composite sample consisting of 15 leaves, here instead is indicated a number of 6 leaves per biological replicate... so?
The authors’ Answer: Thank you for the reviewer’s comment. We are sorry for this statement mistake. In this sentence, we actually mean six leaf mixed samples were used for transcriptome analysis, of which three mixed samples from WT and three from MT. Each mixed sample included 15 leaves and represent one biological replicate. The six mixed samples were named WT#1, WT#2, WT#3, MT#1, MT#2, and MT#3, respectively. We have revised this sentence to ‘total RNAs from WT and MT leaves were sequenced with three biological replicates per variety.’ in the revised manuscript (line 239-240).
Line 197: was only the total mRNA sequenced? Or was all the RNA sequenced and then a filter was applied to remove the rRNA?
The authors’ Answer: Thank you for the reviewer’s comment. We are sorry for this statement mistake. In fact, the total RNA was sequenced and then a filter was applied to remove the rRNA. We have revised this sentence to ‘total RNA from WT and MT leaves were sequenced with three biological replicates per variety’ in the revised manuscript’ in the revised manuscript (line 239-240).
The revisions in the revised manuscript are listed as follows (including the revision according to another reviewer’ s comment):
- Line 21-24: According to the reviewer’s comment, we display the full name of abbreviation in ‘Abstract’ of the revised manuscript.
- Line 25-29: According to the reviewer’s comment, we deleted gene names and added the numbers of genes in ‘Abstract’ of the revised manuscript.
- Line 42: According to the reviewer’s comment, we added a reference [2] in the revised manuscript.
- Line 57-66: According to the reviewer’s comment, we added more research progress about the citrus studies under drought conditions in the revised manuscript.
- Line 66-68: We deleted this sentence in the revised manuscript.
- Line 69: We used ‘aerial organs’ to replace ‘fruits and leaves’ in the revised manuscript.
- Line 79-80: According to the reviewer’s comment, we added these sentence in the revised manuscript.
- Line 80-81: we added the words ‘in fact’ in the revised manuscript.
- Line 86: we added the words ‘due to the lack of wax-related mutant’ in the revised manuscript.
- Line 91-95: According to the reviewer’s comment, we added these sentences to explained the important role of MT to study the relationship between cuticular waxes and citrus drought tolerance in the revised manuscript.
- Line 100: According to the reviewer’s comment, we used ‘real-time quantitative PCR (qRT-PCR)’ to replace ‘real-time quantitative RT-PCR’ in the revised manuscript.
- Line 106: According to the reviewer’s comment, we deleted this sentence.
- Line 109-110: According to the reviewer’s comment, we added the comparison of leaf fresh and dry weight between WT and WT plants.
- Line 112, 113, 115: we changed the figure citing due to adding the bar graph of leaf fresh (H) and dry weight (I) to Figure 1.
- Line 125: we added the sentence of ‘(H) and (I) The fresh and dry weight in WT and MT leaves.’ To the caption of Figure 1.
- Line 131: we added the full name of SEM.
- Line 155-157: According to the reviewer’s comment, we added these sentences to compare the wax composition between WT and MT plants.
- Line 158-168: According to the reviewer’s comment, we deleted these technically description in the revised manuscript.
- Line 168: we added word ‘However’ in this sentence.
- Line 187, 190, 191, 201, 202, 207, 209: According to the reviewer’s comment, we added the full name of MDA, H2O2, SOD, POD, CAT, Pn, Gs, Ci, Tr, WUEi and WUE in the revised manuscript.
- Line 222-236: According to the reviewer’s comment, we deleted this section in the revised manuscript.
- Line 237: we used ‘2.4’ to replace ‘2.5’ in the revised manuscript.
- Line 238-240: we added this sentence in the revised manuscript.
- Line 263-264: we added this sentence in the revised manuscript.
- Line 268-269: we revised the captions of Figure 7.
- Line 272-276: we transferred this paragraph to line 259-263 in the revised manuscript.
- Line 283-285: we this sentence in the revised manuscript.
- Line 295: we used ‘2.5’ to replace ‘2.6’ in the revised manuscript.
- Line 308: According to the reviewer’s comment, we added the full name of ROS in the revised manuscript.
- Line 313-328: According to the reviewer’s comment, we deleted these sentences to avoid enumerations.
- Line 329-333: According to the reviewer’s comment, we added KEGG classification analysis in DEGs of MAPK signaling pathway-plant in the revised manuscript.
- Line 336-337: According to the reviewer’s comment, we deleted the data in brackets.
- Line 337-347: According to the reviewer’s comment, we deleted these sentences to avoid enumerations.
- Line 347-349: According to the reviewer’s comment, we added this sentence in the revised manuscript.
- Line 357-361: we revised the caption of Figure 8 in the revised manuscript.
- Line 363-364: we added the caption of Figure 9 in the revised manuscript.
- Line 370: According to the reviewer’s comment, we deleted the data in brackets.
- Line 371-373: we added this sentence to state the potential role of DEGs encoding transcription factor in the revised manuscript.
- Line 373-381: According to the reviewer’s comment, we deleted these sentences to avoid enumerations.
- Line 382-386: According to the reviewer’s comment, we added KEGG classification analysis in DEGs encoding transcription factor in the revised manuscript.
- Line 417-432: we transferred this paragraph to line 392-407 in the revised manuscript.
- Line 446: we added the words of ‘Further study’ in this sentence.
- Line 442-446, 458-463, 472-473, 514-517,524-526, 528-529, 533-535, 538-543,563-566, 621-625, 630-635, 665-667: According to the reviewer’s comment, we deleted these sentences to shortcut the ‘Discussion’ section.
- Line 490-491: we deleted the full name of ROS.
- Line 506: According to the reviewer’s comment, we use ‘CO2’ to replace ‘CO2’ in this sentence.
- Line 610-612, 642: According to the reviewer’s comment, we deleted the data in brackets.
- Line 629-630, 642-644, 654-656, 671-681: According to the reviewer’s comment, we added some discussion mainly in molecular mechanism part in the revised manuscript.
- Line 692-695: we added the seeds origin and nutritional soil formula in this sentence.
- Line 698: we used ‘rootstocks’ to replace ‘stocks’.
- Line 700 to 705: According to the reviewer’s comment, we put the method of leaf sampling in here.
- Line 706: we added the word ‘fresh and dry weight’ in this sentence.
- Line 709-719: According to the reviewer’s comment, we added the parameters and units detected by CR-300 device and the method to determine fresh and dry weight.
- Line 736: we deleted the full name of SEM and the word ‘analysis’.
- Line 776-777: we deleted the full name of Pn, Gs, Ci and Tr.
- Line 778-781, 785-791, 794-853: According to the reviewer’s comment, we added the methods to determine ion leakage, the contents of MDA, proline, soluble sugar, chlorophyll, H2O2, the activities of SOD, POD and CAT, stomatal conductance, intercellular CO2 concentration and transpiration rate.
- Line 884: we added the full name of ORFs.
- Line 890: we deleted the full name of DEGs.
- Line 892-893: we deleted the Pearson correlation analysis and PCA methods due to the deletion of Pearson correlation analysis and PCA analysis according to the reviewer’s comment.
- Line 895-896: we deleted the full name of GO and KEGG.
Changes in ‘Supplementary Materials’:
- Line 931-937: According to the reviewer’s comment, we deleted Figure S1, Figure S2, Table S1 to Table S5 and their captions.
- Line 943-947: According to the reviewer’s comment, we added two supplementary tables (Table S7 and (Table S9) and their captions.
Changes in supplementary ‘References’
- Line 965-966, 988-997, 1030-1032, 1190-1212: we added 15 references in the revised manuscript.
- Line 1044-1045, 1049-1050, 1075-1076, 1082-1083: we deleted 4 references in the revised manuscript.
- Line 1173-1174: we transferred this reference to to line 1169-1170 in the revised manuscript.
Changes in supplementary Figures
- Figure 1: We added fresh (H) and dry weight (I) in Figure 1.
- Figure 2: We also increased the resolution from 600 dpi to 1000 dpi to improve the figure quality.
- Figure 7: The (B) and (C) were exchanged.
- Figure 8: According to the reviewer’s comment, we have rebuilt the Heatmap based on the log2(FPKM+1) values of six WT and MT samples.
- Figure 9: According to the reviewer’s comment, we added Figure 9 to perform KEGG classification analysis for DEGs in MAPK signaling pathway-plant and DEGs encoding transcription factor.

Reviewer 2 Report
The work entitled 'Transcriptome and physiological analyses of a navel orange mutant with improved drought tolerance and water use efficiency caused by increases of cuticular wax accumulation and ROS scavenging capacity' shows the use of a mutant to analyze the relevance of waxes and other mechanisms of drought tolerance. The results are consistent and very interesting, however, the organization can be confusing, many data saturate the reader and are not totally necessary, and the way of providing the information is more reminiscent of a report than of a scientific article in some moments. I would recommend a reorganization of the text to support the reader and thus improve the dissemination of the relevant results of this text. The use of a new mutant is the best contribution of this publication and could open the door to use it as a model for other stresses. A very correct and interesting work. Here you can find some comments:
General:
-Despite the relevance of the new mutant, seems not very well explained the implication of such mutation, and by moments in the text, is not clear why is relevant in this study, when is clearly the best input to analyze drought events
Abstract
-Not sure if including the abbreviated names is recommended or not in the abstract. As recommendations, display first the full name the first time they appear in the text
-Maybe not necessary to include all genes tested here. Save them for material and methods or results
Introduction
-Last paragraph, talking about critics, need to be re-organized. Like this could be confusing and seems not to have a structure
-I think the citrus studies under drought conditions are very underrepresented. Consider to include more state-of-the-art about the topic
Results
-Again, first time mentioning abbreviated concepts, please write first the full version
-Phenotyping:
Apart from visual comparison, a weight quantification (dry weight) would be necessary as an indicator
-Comparison of wax:
I would recommend not to expose technically the components, but try to include as a comparison to make the reading more meaningful
-Comparative transcriptome:
This section is adding no value to the manuscript
-All section of transcriptomic analyses could be improved to make it easier to follow, but also to avoid reiterates and enumerations (this is not useful)
-Figure 8: Heatmap is not well-built, Each column is consisting in one sample? The mean of three? I don't think is the best way to express. Also consider to re-organize by GO or functional groups to help in with the result interpretation
-Sections 2.6 and 2.7:
The data in brackets as "(84 DEGs)" or "(8 downregulated)", are completely meaningless, it's better if you compare these numbers and give some context or relevance/importance of such a value
Discussion:
-Despite is correct, maybe consider shortcutting, it's very dense and this can create confusion
-Dedicate very long time for some aspects, meanwhile some others are merely mentioned, despite their relevance. I would balance this section
Material and Methods:
-Section 4.2: Include the parameters and units detected by CR-300 device
-CO2 with subscript: 2
Supplementary material: Websites for support are not working (as, www.mdpi.com/xxx/s1). If this is just editing issue is ok, but have a look
Author Response
Dear reviewer:
Thank you very much for your comment. We revised the manuscript in accordance with the reviewers’ comments, and carefully proof-read the manuscript to minimize typographical, grammatical, and bibliographical errors. Here below is our description on revision according to the reviewers’ comments.
Point-by-point answers to reviewer #2:
The work entitled 'Transcriptome and physiological analyses of a navel orange mutant with improved drought tolerance and water use efficiency caused by increases of cuticular wax accumulation and ROS scavenging capacity' shows the use of a mutant to analyze the relevance of waxes and other mechanisms of drought tolerance. The results are consistent and very interesting, however, the organization can be confusing, many data saturate the reader and are not totally necessary, and the way of providing the information is more reminiscent of a report than of a scientific article in some moments. I would recommend a reorganization of the text to support the reader and thus improve the dissemination of the relevant results of this text. The use of a new mutant is the best contribution of this publication and could open the door to use it as a model for other stresses. A very correct and interesting work. Here you can find some comments:
General:
-Despite the relevance of the new mutant, seems not very well explained the implication of such mutation, and by moments in the text, is not clear why is relevant in this study, when is clearly the best input to analyze drought events
The authors’ Answer: Thank you for the reviewer’s comment, we have explained the important role of MT to study the relationship between cuticular waxes and citrus drought tolerance in the revised manuscript (line 87-95).
Abstract
-Not sure if including the abbreviated names is recommended or not in the abstract. As recommendations, display first the full name the first time they appear in the text
The authors’ Answer: According to the reviewer’s comment, we display the full name of abbreviation in ‘Abstract’ of the revised manuscript (line 21-24).
-Maybe not necessary to include all genes tested here. Save them for material and methods or results
The authors’ Answer: According to the reviewer’s comment, we deleted gene names and added the numbers of genes in ‘Abstract’ of the revised manuscript (line 25-29).
Introduction
-Last paragraph, talking about critics, need to be re-organized. Like this could be confusing and seems not to have a structure
The authors’ Answer: According to the reviewer’s comment, we have reorganized the Last paragraph in the revised manuscript (line 57-95).
-I think the citrus studies under drought conditions are very underrepresented. Consider to include more state-of-the-art about the topic
The authors’ Answer: According to the reviewer’s comment, we added more research progress about the citrus studies under drought conditions in the revised manuscript (line 57-66).
Results
-Again, first time mentioning abbreviated concepts, please write first the full version
The authors’ Answer: According to the reviewer’s comment, we have written all the full names of abbreviation when first mentioned in the revised manuscript (line 131, 187, 190, 191, 201, 202, 207, 209, 308, 885).
-Phenotyping:
Apart from visual comparison, a weight quantification (dry weight) would be necessary as an indicator
The authors’ Answer: Thank you for the reviewer’s comment, we have added the leaf fresh and dry weight of WT and MT plants in Figure 1H,I and state this results in the revised manuscript (line 109, 110). The method to determine leaf fresh and dry weight of WT and MT plants were added in in the revised manuscript (line 716-720).
-Comparison of wax:
I would recommend not to expose technically the components, but try to include as a comparison to make the reading more meaningful
The authors’ Answer: According to the reviewer’s comment, we have used a comparison of wax components to replace the technically description in the revised manuscript (line 155-157).
-Comparative transcriptome:
This section is adding no value to the manuscript
The authors’ Answer: According to the reviewer’s comment, we have deleted this section in the revised manuscript (line 222-236).
-All section of transcriptomic analyses could be improved to make it easier to follow, but also to avoid reiterates and enumerations (this is not useful)
The authors’ Answer: According to the reviewer’s comment, we have improved the transcriptomic analyses and deleted some reiterates and enumerations in the revised manuscript (line 238-390).
-Figure 8: Heatmap is not well-built, each column is consisting in one sample? The mean of three? I don't think is the best way to express. Also consider to re-organize by GO or functional groups to help in with the result interpretation
The authors’ Answer: According to the reviewer’s comment, we have rebuilt the Heatmap based on the log2(FPKM+1) values of six WT and MT samples (see Figure 8 in the revised manuscript). In addition, we also performed KEGG classification analysis for DEGs in MAPK signaling pathway-plant and DEGs encoding transcription factor (Figure 9 in the revised manuscript).
-Sections 2.6 and 2.7:
The data in brackets as "(84 DEGs)" or "(8 downregulated)", are completely meaningless, it's better if you compare these numbers and give some context or relevance/importance of such a value
The authors’ Answer: According to the reviewer’s comment, we have deleted the data in brackets and added some description about the importance of the enriched pathway and transcription factor families in the revised manuscript (line 313-349, 372-387).
Discussion:
-Despite is correct, maybe consider shortcutting, it's very dense and this can create confusion
The authors’ Answer: According to the reviewer’s comment, we have shortcut the discussion part in the revised manuscript (line 435-690).
-Dedicate very long time for some aspects, meanwhile some others are merely mentioned, despite their relevance. I would balance this section
The authors’ Answer: According to the reviewer’s comment, we have deleted some repeat and meaningless part in discussion (line 442-446, 458-463, 472-473, 514-517,524-526, 528-529, 533-535, 538-543,563-566, 621-625, 630-635, 665-667), and added some discussion mainly in molecular mechanism part (line 629-630, 642-644, 654-656, 671-681) in the revised manuscript.
Material and Methods:
-Section 4.2: Include the parameters and units detected by CR-300 device
The authors’ Answer: According to the reviewer’s comment, we have added the parameters and units detected by CR-300 device in the revised manuscript (line 710-716).
-CO2 with subscript: 2
The authors’ Answer: Thank you for the reviewer’s comment, we are very sorry for this maitake and used subscript 2 in CO2 throughout the revised manuscript.
Supplementary material: Websites for support are not working (as, www.mdpi.com/xxx/s1). If this is just editing issue is ok, but have a look
The authors’ Answer: Thank you for the reviewer’s comment, this is the template in this journal, I think the websites will be activated after publication.
The revisions in the revised manuscript are listed as follows (including the revision according to another reviewer’ s comment):
- Line 21-24: According to the reviewer’s comment, we display the full name of abbreviation in ‘Abstract’ of the revised manuscript.
- Line 25-29: According to the reviewer’s comment, we deleted gene names and added the numbers of genes in ‘Abstract’ of the revised manuscript.
- Line 42: According to the reviewer’s comment, we added a reference [2] in the revised manuscript.
- Line 57-66: According to the reviewer’s comment, we added more research progress about the citrus studies under drought conditions in the revised manuscript.
- Line 66-68: We deleted this sentence in the revised manuscript.
- Line 69: We used ‘aerial organs’ to replace ‘fruits and leaves’ in the revised manuscript.
- Line 79-80: According to the reviewer’s comment, we added these sentence in the revised manuscript.
- Line 80-81: we added the words ‘in fact’ in the revised manuscript.
- Line 86: we added the words ‘due to the lack of wax-related mutant’ in the revised manuscript.
- Line 91-95: According to the reviewer’s comment, we added these sentences to explained the important role of MT to study the relationship between cuticular waxes and citrus drought tolerance in the revised manuscript.
- Line 100: According to the reviewer’s comment, we used ‘real-time quantitative PCR (qRT-PCR)’ to replace ‘real-time quantitative RT-PCR’ in the revised manuscript.
- Line 106: According to the reviewer’s comment, we deleted this sentence.
- Line 109-110: According to the reviewer’s comment, we added the comparison of leaf fresh and dry weight between WT and WT plants.
- Line 112, 113, 115: we changed the figure citing due to adding the bar graph of leaf fresh (H) and dry weight (I) to Figure 1.
- Line 125: we added the sentence of ‘(H) and (I) The fresh and dry weight in WT and MT leaves.’ To the caption of Figure 1.
- Line 131: we added the full name of SEM.
- Line 155-157: According to the reviewer’s comment, we added these sentences to compare the wax composition between WT and MT plants.
- Line 158-168: According to the reviewer’s comment, we deleted these technically description in the revised manuscript.
- Line 168: we added word ‘However’ in this sentence.
- Line 187, 190, 191, 201, 202, 207, 209: According to the reviewer’s comment, we added the full name of MDA, H2O2, SOD, POD, CAT, Pn, Gs, Ci, Tr, WUEi and WUE in the revised manuscript.
- Line 222-236: According to the reviewer’s comment, we deleted this section in the revised manuscript.
- Line 237: we used ‘2.4’ to replace ‘2.5’ in the revised manuscript.
- Line 238-240: we added this sentence in the revised manuscript.
- Line 263-264: we added this sentence in the revised manuscript.
- Line 268-269: we revised the captions of Figure 7.
- Line 272-276: we transferred this paragraph to line 259-263 in the revised manuscript.
- Line 283-285: we this sentence in the revised manuscript.
- Line 295: we used ‘2.5’ to replace ‘2.6’ in the revised manuscript.
- Line 308: According to the reviewer’s comment, we added the full name of ROS in the revised manuscript.
- Line 313-328: According to the reviewer’s comment, we deleted these sentences to avoid enumerations.
- Line 329-333: According to the reviewer’s comment, we added KEGG classification analysis in DEGs of MAPK signaling pathway-plant in the revised manuscript.
- Line 336-337: According to the reviewer’s comment, we deleted the data in brackets.
- Line 337-347: According to the reviewer’s comment, we deleted these sentences to avoid enumerations.
- Line 347-349: According to the reviewer’s comment, we added this sentence in the revised manuscript.
- Line 357-361: we revised the caption of Figure 8 in the revised manuscript.
- Line 363-364: we added the caption of Figure 9 in the revised manuscript.
- Line 370: According to the reviewer’s comment, we deleted the data in brackets.
- Line 371-373: we added this sentence to state the potential role of DEGs encoding transcription factor in the revised manuscript.
- Line 373-381: According to the reviewer’s comment, we deleted these sentences to avoid enumerations.
- Line 382-386: According to the reviewer’s comment, we added KEGG classification analysis in DEGs encoding transcription factor in the revised manuscript.
- Line 417-432: we transferred this paragraph to line 392-407 in the revised manuscript.
- Line 446: we added the words of ‘Further study’ in this sentence.
- Line 442-446, 458-463, 472-473, 514-517,524-526, 528-529, 533-535, 538-543,563-566, 621-625, 630-635, 665-667: According to the reviewer’s comment, we deleted these sentences to shortcut the ‘Discussion’ section.
- Line 490-491: we deleted the full name of ROS.
- Line 506: According to the reviewer’s comment, we use ‘CO2’ to replace ‘CO2’ in this sentence.
- Line 610-612, 642: According to the reviewer’s comment, we deleted the data in brackets.
- Line 629-630, 642-644, 654-656, 671-681: According to the reviewer’s comment, we added some discussion mainly in molecular mechanism part in the revised manuscript.
- Line 692-695: we added the seeds origin and nutritional soil formula in this sentence.
- Line 698: we used ‘rootstocks’ to replace ‘stocks’.
- Line 700 to 705: According to the reviewer’s comment, we put the method of leaf sampling in here.
- Line 706: we added the word ‘fresh and dry weight’ in this sentence.
- Line 709-719: According to the reviewer’s comment, we added the parameters and units detected by CR-300 device and the method to determine fresh and dry weight.
- Line 736: we deleted the full name of SEM and the word ‘analysis’.
- Line 776-777: we deleted the full name of Pn, Gs, Ci and Tr.
- Line 778-781, 785-791, 794-853: According to the reviewer’s comment, we added the methods to determine ion leakage, the contents of MDA, proline, soluble sugar, chlorophyll, H2O2, the activities of SOD, POD and CAT, stomatal conductance, intercellular CO2 concentration and transpiration rate.
- Line 884: we added the full name of ORFs.
- Line 890: we deleted the full name of DEGs.
- Line 892-893: we deleted the Pearson correlation analysis and PCA methods due to the deletion of Pearson correlation analysis and PCA analysis according to the reviewer’s comment.
- Line 895-896: we deleted the full name of GO and KEGG.
Changes in ‘Supplementary Materials’:
- Line 931-937: According to the reviewer’s comment, we deleted Figure S1, Figure S2, Table S1 to Table S5 and their captions.
- Line 943-947: According to the reviewer’s comment, we added two supplementary tables (Table S7 and (Table S9) and their captions.
Changes in supplementary ‘References’
- Line 965-966, 988-997, 1030-1032, 1190-1212: we added 15 references in the revised manuscript.
- Line 1044-1045, 1049-1050, 1075-1076, 1082-1083: we deleted 4 references in the revised manuscript.
- Line 1173-1174: we transferred this reference to to line 1169-1170 in the revised manuscript.
Changes in supplementary Figures
- Figure 1: We added fresh (H) and dry weight (I) in Figure 1.
- Figure 2: We also increased the resolution from 600 dpi to 1000 dpi to improve the figure quality.
- Figure 7: The (B) and (C) were exchanged.
- Figure 8: According to the reviewer’s comment, we have rebuilt the Heatmap based on the log2(FPKM+1) values of six WT and MT samples.
- Figure 9: According to the reviewer’s comment, we added Figure 9 to perform KEGG classification analysis for DEGs in MAPK signaling pathway-plant and DEGs encoding transcription factor.

Round 2
Reviewer 1 Report
I thank the authors for answering all questions, and revised the text according to the revisions made